# RobustBlack: Challenging Black-Box Adversarial Attacks on State-of-the-Art Defenses

## Abstract

Although adversarial robustness has been extensively studied in white-box settings, recent advances in black-box attacks (including transfer- and query-based approaches) are primarily benchmarked against weak defenses, leaving a significant gap in the evaluation of their effectiveness against more recent and moderate robust models (e.g., those featured in the Robustbench leaderboard). In this paper, we question this lack of attention from black-box attacks to robust models. We benchmark the effectiveness of recent black-box attacks against both top-performing and standard defense mechanisms, on the ImageNet dataset. Our empirical evaluation reveals the following key findings: (1) the most advanced black-box attacks struggle to succeed even against simple adversarially trained models; (2) robust models that are optimized to withstand strong white-box attacks, such as AutoAttack, also exhibit enhanced resilience against black-box attacks; and (3) robustness alignment between the surrogate models and the target model plays can significantly impact the success rate of transfer-based attacks.

## 1 Introduction

Since the discovery that deep learning models are susceptible to minor input disturbances, resulting in adversarial examples (AE) (Goodfellow et al., 2015), the development of robust models has become one of the most active topics in the machine learning community. This topic has been thoroughly explored for white-box settings, leading to standardized attacks (Croce & Hein, 2020b) and benchmarks (Croce et al., 2020).

However, realistic evaluations of adversarial attacks have not yet been explored. It involves scenarios where the attacker has limited or no knowledge of the target model to be attacked. These scenarios are referred to as black-box (or gray-box with partial knowledge (Guo et al., 2018)). Black-box attacks belong to one of the two following settings: When the output of the model provides both category labels and scores, it is referred to as a *score-based* black-box attack. In contrast, if the output includes only category labels, it is named a *decision-based* black-box attack (Brendel et al., 2018). Given that many computer vision Application Programming Interface (APIs) (Google, 2023; Imagga, 2023) provide image prediction categories alongside scores, most research in a black-box setting considers score-based black-box attacks. Hence, this work focuses on this category.

State-of-the-art (SOTA) black-box attacks leverage one or both of the following mechanisms: (1) adversarial examples transferability and (2) iterative queries with meta-heuristics. Transferability methods craft adversarial examples on a specifically built surrogate model (with white-box access) such that this example should transfer to (i.e. also fools) a target model (to which they have black-box access). Iterative query methods use query feedback from the target model to guide or refine the optimization process to uncover its vulnerabilities.

Transferability is difficult to achieve when the target differs from the surrogate (in terms of architecture, training, or dataset) and the factors behind transferability (or lack thereof) remain an active field of study (Charles et al., 2020; Dong et al., 2018a; Li et al., 2018; Wu et al., 2020a). This is because adversarial attacks crafted on the surrogate lead to examples that maximize the loss function of the surrogate model

(Goodfellow et al., 2015; Kurakin et al., 2017), while the target model has a different loss function. Methods to improve transferability mostly rely on building diversity during the optimization of adversarial examples (Li et al., 2018; Wu et al., 2020a; Xie et al., 2019a).

Traditional iterative-query attacks also face multiple challenges: With increased search space (large images, for example), common query attacks show a significant decrease in the Attack Success Rate (ASR) or require large amounts of queries to achieve sufficient ASR (Mohaghegh Dolatabadi et al., 2020; Huang & Zhang, 2020; Feng et al., 2022). BASES (Cai et al., 2022b), a recent Black-box Surrogate Ensemble Search attack achieved however more than 95% ASR with a few queries ($<5$) on average across a large scope of non-robust models.

Though the initial motivation for black-box attacks and their evaluation protocol is to improve the realism of robustness evaluations, they generally suffer from a common pitfall: they do not consider robust models, which have already demonstrated their improved performance against white-box attacks. We hypothesize that the evaluation results of black-box attacks can be misleading because even simple robustification mechanisms can be sufficient to successfully evade black-box attacks. Going further, we want to assess whether innovations in defenses provide positive benefits against black-box attacks like they did in white-box settings.

To the best of our knowledge, this paper is the first to study the effectiveness of black-box attacks against strandardized defenses to demonstrate the need to confront black-box attacks to existing defenses.

**Our contributions can be summarized as follows:**

1. We demonstrate that simple adversarial training mechanisms reduce the effectiveness of black-box attacks proven effective against standard models, underscoring the need for more advanced black-box attack strategies to address the robustness of real-world models and systems.

2. We show that white-box robustness could serve as a proxy for black-box robustness. Defenses optimized against AutoAttack generalize well to black-box scenarios.

3. We demonstrate that these effective defense mechanisms can inadvertently contribute to enhancing the success rate of black-box attacks. By using robust models as surrogates, attacks can generate adversarial examples more likely to transfer to robust models, leading to an average increase in success rate of 6.49 percentage points across attacks and target models.

Our contributions send a clear message to the adversarial research community on the importance of considering white-box defenses in the evaluation of black-box attacks. They also urge researchers working on defense mechanisms to study how their defenses can help improve transfer-based attacks.

## 2 Background

### 2.1 Preliminaries

**Adversarial perturbation Croce et al. (2020):** Let $x \in \mathbb{R}^d$ be an input point and $y \in \{1, \ldots, C\}$ be its correct label. For a classifier $f : \mathbb{R}^d \to \mathbb{R}^C$, we define a *successful adversarial perturbation* with respect to the perturbation set $\Delta \subseteq \mathbb{R}^d$ as a vector $\vec{\delta} \in \mathbb{R}^d$ such that

$$\arg\max_{c \in \{1,\ldots,C\}} f(x + \delta)_c \neq y \quad \text{and} \quad \delta \in \Delta,$$

where the perturbation set $\Delta$ is chosen such that all examples in $x + \delta$ have $y$ as their true label. This motivates a robustness measure called *robust accuracy*, which is the fraction of data points on which the classifier $f$ predicts the correct class for all possible perturbations from the set $\Delta$. Computing the exact Robust Accuracy (RA) is in general intractable and, when considering $\ell_p$-balls as $\Delta$, NP-hard even for single-layer neural networks. In practice, an *upper bound* for robust accuracy is determined by *adversarial attacks*,

generally involving optimization of a differentiable loss function or a reward through search algorithms that aim to identify a successful adversarial perturbation. The tightness of the upper bound depends on the strength of the attack.

**Success rate:** Given that different models show varying clean performances (eg, test accuracy on the original set), robust accuracy will be impacted by the initial performance as much as the intrinsic robustness of the models. Thus, we base our study on an agnostic test performance metric: attack success rate (ASR). We define ASR for a classifier $f$ under attack $\mathcal{A}; \mathcal{A}(x) = x + \delta$ as:

$$\text{ASR}(f, \mathcal{A}) = \mathbb{E}_{(x,y)\sim\mathcal{D}}\left[\mathbb{I}\left(f(\mathcal{A}(x)) \neq f(x)\right)\right]$$

For a model with a 100% test accuracy, the two metrics are related $\text{ASR} = 1 - \text{RA}$.

### 2.2 Black-Box Attacks: Transfer-Based and Query-Based Methods

Black-box attacks have seen notable advancements, especially within transfer-based and query-based approaches. Transfer-based attacks rely on adversarial examples generated by a surrogate model, which are then tested on a target model. Early approaches like Projected Gradient Descent (PGD) (Madry et al., 2017) laid the groundwork for such attacks by iteratively optimizing perturbations to maximize the success rate on the target model. This foundational method was enhanced by the Momentum Iterative Fast Gradient Sign Method (MI-FGSM), which incorporated momentum into PGD to improve transferability and stability across iterations (Dong et al., 2018b). Further innovations, such as the Diverse Input Fast Gradient Sign Method (DI-FGSM) and Translation-Invariant FGSM (TI-FGSM), introduced input transformations and image translations, respectively, to increase the robustness and generalizability of adversarial examples across different models (Xie et al., 2019b; Dong et al., 2019).

Subsequent methods such as Variance Tuning (VMI, VNI) refined gradient calculations considering gradient variance between iterations (Wang & He, 2021), leading to more effective adversarial perturbations. Similarly, ADMIX leveraged diverse inputs by mixing the target image with randomly sampled images to generate more transferable attacks (Wang et al., 2021). The Skip Gradient Method (SGM) focused on model architecture, using skip connections which facilitate the creation of adversarial examples that have high transferability (Wu et al., 2020b). Universal Adversarial Perturbations (UAP) took a different approach by focusing on universality, generating a single small perturbation that can effectively disrupt a classifier in most natural images (Moosavi-Dezfooli et al., 2017). In particular, Ghost networks (GHOST) and Large Geometric Vicinity (LGV) contributed to transferability by modifying skip connections in surrogate models and by exploring variations in surrogate models' vicinity, respectively (Li et al., 2020; Gubri et al., 2022).

In query-based attacks, which generally have higher success rates than transfer attacks (Andriushchenko et al., 2019), methods like Zeroth Order Optimization (ZOO), Decision Boundary Attack (DBA), Natural Evolutionary Strategies (NES),and HopSkipJump rely on iterative query feedback to optimize perturbations, often not utilizing surrogate models to boost efficiency (Brendel et al., 2018; Chen et al., 2017; Ilyas et al., 2018; Chen et al., 2020).

The following attacks then focused on reducing the number of queries. Andriushchenko et al. (2019) introduced a query-efficient black-box attack using randomized square-shaped updates at image boundaries, SimBA (Guo et al., 2019) proposed a simple attack using orthogonal search directions (e.g., DCT basis). Sign-OPT (Cheng et al., 2020) introduces a hard-label attack estimating gradient signs instead of magnitudes, while SignHunt (Al-Dujaili & O'Reilly, 2020) uses sign bits for gradient estimation. to minimize the number of queries. RayS (Chen & Gu, 2020) reformulates the boundary search as discrete optimization, eliminating gradient estimation.

Recent work combined both surrogate representations and query feedback to create highly effective and efficient black-box attacks. Representative attacks include Transferable Model-based Embedding (TREMBA) and BASES (Huang & Zhang, 2019; Cai et al., 2022a). BASES leverages an ensemble of surrogates with query feedback to dynamically adjust surrogate weighting, while TREMBA utilizes a generator trained with surrogate models, exploring latent space for more effective query attacks.

## 2.3 Defenses Against Strong Adversarial Attacks

In parallel, adversarial defenses have evolved to mitigate the risks posed by increasingly sophisticated white-box attacks. As defenses became more robust, the need for a reliable, standardized benchmark to accurately assess their effectiveness grew increasingly apparent. In response, AutoAttack emerged as a reliable, parameter-free method to evaluate adversarial robustness, offering a computationally affordable and a standardized benchmark applicable to various models (Croce & Hein, 2020b). Building on this, Robustbench established a leaderboard specifically to evaluate adversarial robustness, as outlined by (Croce et al., 2020). This leaderboard includes results on ImageNet large-scale datasets (Deng et al., 2009), and offers a ranking comparison of various defense strategies in specific perturbation budgets. Among the defenses featured on the leaderboard are those proposed by (Salman et al., 2020; Singh et al., 2024; Liu et al., 2024; Bai et al., 2024; Madry et al., 2017), which offers a thorough assessment of the effectiveness of the leading defense mechanisms. Traditional defenses, such as Madry's adversarial training, initially combined clean and adversarial data during training, establishing a baseline for robustness (Madry et al., 2017). Recent advancements have incorporated complex pre-training schemes and data augmentations (e.g., ConvStem architectures, RandAugment, MixUp, and CutMix). Singh et al. (2024) made alterations to the convnets architecture, particularly substituting PatchStem with ConvStem, and modified the training scheme to enhance the robustness against previously unencountered l1 and l2 threat models. Liu et al. (2024) further advances the training scheme with large-scale pre-training, combined with label smoothing, weight decay, and Exponential Moving Averaging (EMA) to enhance generalization against unseen adversarial examples. Bai et al. (2024) builds on a given defense framework (Liu et al., 2024), focusing specifically on balancing clean and robust accuracy through the nonlinear mixing of robust and standard models, addressing the often-seen tradeoff between clean and robust performance.

## 2.4 Robustness Under Black-Box Setting

In this part, we focus on how our work introduces a novel contribution compared to previous studies on defenses against black-box adversarial attacks in image classification tasks. We discuss five representative works (Papernot et al., 2017; Dong et al., 2020; Mahmood et al., 2021a;b; Ghaffari Laleh et al., 2022), highlighting their key contributions and how our approach builds on or complements these efforts by addressing gaps and introducing new perspectives in this domain.

Papernot et al. (2017) introduced a black-box attack strategy targeting deep neural network (DNN) models and evaluated its effectiveness against adversarial training and defensive distillation. They varied the magnitude of the perturbation used during training and the attack phase and demonstrated that small perturbations of adversarial training in training led to gradient masking, which their attack could bypass by doubling the perturbation budget in the attacking phase, whereas larger perturbations in training improved robustness. In contrast, our work focuses on evaluating adversarially trained SOTA defenses against black-box attacks using a fixed small perturbation 4 over 255 in the attack phase.

Dong et al. (2020) evaluated adversarial robustness in image classification tasks against white-box and black-box attacks. Their work addressed various defense techniques such as robust training, input transformation, randomization, certified defenses, and model ensembling. Our study seeks to complement this by focusing specifically on SOTA robust training-based defenses. Moreover, our work explores a broader landscape of black-box attacks by including recent methods such as ADMIX, BASES, and TREMBA. In doing so, we provide an analysis of the leading robust training defenses against recent black-box adversarial strategies.

Mahmood et al. (2021a) noted that the majority existing defenses primarily address white-box attacks, neglecting the crucial aspect of black-box adversarial robustness. They provided a wide evaluation of adversarial defenses with a Convolutional Neural Network (CNN) architecture on benchmark datasets such as CIFAR-10 (Krizhevsky, 2009) and Fashion-MNIST (Xiao et al., 2017) against 12 attacks. In a follow-up work, Mahmood et al. (2021b) extended this investigation to Vision Transformers (ViTs), evaluating their adversarial robustness against white-box attacks and two black-box attacks on ImageNet. They suggested that ensemble defenses can enhance robustness when attackers lack access to model gradients. Additionally, They examined the transferability of adversarial examples between CNNs and transformers. Building upon and complementing these efforts, we conduct our evaluations on ImageNet, incorporating both CNN and

transformer architectures, and including an ensemble defense strategy that combines robust and standard base models. Our approach further investigates transferability between adversarially trained and standardly trained models. We leverage SOTA robust models to challenge the attacks used to evaluate robustness.

In the field of computational pathology, Ghaffari Laleh et al. (2022) examined adversarial robustness with a focus on CNN and transformer models. This work addressed certain adversarial attacks (including Fast Adaptive Boundary (FAB) (Croce & Hein, 2020a), and Square (Andriushchenko et al., 2020)), as well as adversarial training using (PGD) with Dual-Batch Normalization (DBN). Our work extends this line of inquiry by incorporating adversarially trained SOTA models and a more comprehensive suite of black-box attacks, providing additional context for these attacks by increasing their number and offering a more challenging confrontation against SOTA defenses.

A recent line of research investigated pre-processing and randomness injection defenses against black-box attacks. Qin et al. (2021) proposes adding Gaussian noise to queries to disrupt gradient estimation in black-box attacks, combining it with Gaussian-augmentation fine-tuning for improved robustness-accuracy trade-offs, Chen et al. (2022) proposed the Adversarial Attack on Attackers (AAA) defense. It introduces output logit perturbation to misdirect score-based attacks while preserving clean accuracy and improving calibration. Nguyen et al. (2024) showed that randomizing hidden features provides better robustness than input randomization against query-based attacks, with plug-and-play implementation and minimal accuracy loss. Sitawarin et al. (2023) demonstrated that unaware preprocessing reduces attack efficacy by $7\times$, and designed preprocessor-aware attacks that easily overcome such defenses.

## 2.5 Explaining Transferability and Robustness in Black-Box Setting

Various research investigated the factors behind the transferability (or lack-off) of adversarial examples. In our study, we covered each theory with at least one representative method.

**Decision Boundary Similarity (Liu et al., 2016).** Models trained on the same task develop similar decision boundaries. Adversarial perturbations are highly aligned with weight vectors across models. Attacks relying on these mechanisms include PGD and HopSkipJump.

**Gradient Similarity (Demontis et al., 2019).** Two main factors contribute to transferability: (1) similarity between gradient directions of source and target models, and (2) smoothness of the loss landscape. Higher gradient similarity and lower variance in loss landscapes lead to increased transferability. Representative attacks include MI-FGSM and Variance Tuning (VMI/VNI).

**Shared Adversarial Subspaces (Tramèr et al., 2017).** Adversarial examples span contiguous subspaces of large dimensionality. When different models are trained on the same task, a fraction of their adversarial subspaces overlap. DI-FGSM, Ghost Networks, and LGV use these mechanisms.

**Non-Robust Features (Ilyas et al., 2019).** Models learn both robust features (perceivable by humans) and non-robust features (imperceptible but statistically correlated with labels). Adversarial examples exploit non-robust features. ADMIX and NES leverage this theory.

**Model Complexity/Capacity (Wu & Zhu, 2020).** Model-specific factors including architecture and capacity influence transferability. Adversarial examples generated from simpler/shallower models tend to transfer better to complex models (e.g. with SGM).

**Interaction-Based (Wang et al., 2020).** There is a negative correlation between adversarial transferability and interactions inside adversarial perturbations. Less interaction within perturbation components leads to higher transferability. TI-FGSM falls within this category.

**Knowledge Transferability (Liang et al., 2021).** Models with high knowledge transfer (e.g., via fine-tuning) exhibit stronger adversarial example transfer. The embeddings trained in TREMBA leverage this knowledge transfer.

**Flatness/Manifold (Fan et al., 2024).** There are conflicting results that higher flatness of adversarial examples enables better cross-model transferability. BASES, through its search algorithm, finds optimal weights to explore the manifold.

Other researchers explored specifically the link between whitebox robustness and transferability. Springer et al. (2021) investigated the link between the robustness of the source models and the robustness of the target models for transferable attacks. Their results show as expected that adversarial examples generated using non-robust networks do not transfer to the adversarially trained networks. However, what they consider as robust source models is a simple Resnet50 with vanilla adversarial training, which has since been shown to only be slightly robust Croce et al. (2020). Our study on the other hand leverages state-of-the-art robustitifcation mechanisms and explores a large set of architecture for robust surrogates and targets. Although our initial results match, our study highlights further insights (e.g., using robust sources is actually detrimental when targeting non-robust targets).

Zhang et al. (2024) confirmed the impact of model smoothness and gradient similarity by exploring the impact of weakly adversarial training (that is, training with mildly perturbed examples). The approach used to train the model, including data augmentation, synthetic data, regularization are considered in some of the models of our study. Our work focuses on robustification mechanism using extreme augmentations. All in all, our results are complementary; as they explore "mildly robust models" and we focus on "extremely robust models".

## 3 Success Rate of Black-Box Attacks Against Adversarial Training

As preliminaries, we investigate the effectiveness of 12 black-box adversarial attacks against Madry adversarial training (Madry et al., 2017) on a single, relatively simple ResNet50 model (He et al., 2016). The purpose of these experiments is to check to what extent black-box attacks, typically successful against vanilla models, remain as successful when confronted to adversarially trained models. This preliminary experiment provides us with the first insight into the impact of defenses studied in white-box settings on the effectiveness of black-box attacks.

### 3.1 Experimental Setup

We evaluated the attack success rate of 11 black-box adversarial attacks against a standardly trained model (vanilla) and an adversarially trained model. We selected a ResNet50 vanilla target and a ResNet50 robust target that underwent adversarial training. We opted for the same model architecture to eliminate the variations that could arise from differing model architectures. We followed the Robustbench evaluation protocol (Croce et al., 2020) and reused the original implementation of each attack. All attacks were untargeted and bound to an $L_\infty$ maximum perturbation with a distance of $4/255$. We evaluated the ASR on 5000 examples from the ImageNet validation set, we used the same image identifiers (IDs) as in RobustBench. All the experiments are repeated with three random seeds.

As outlined in 2.2, we selected nine transfer-based and two query-based adversarial attacks to provide a comprehensive evaluation of adversarial techniques, balancing foundational, and SOTA approaches. For transfer attacks, our selection emphasizes the diversity of techniques by including methods targeting input transformations (TI, DI), gradient optimization (MI, VNI, VMI, ADMIX), universality (UAP), architectural nuances (GHOST), and geometry (LGV). For query-based attacks, we focused on methods that improve query efficiency and performance by leveraging surrogate models, and that rely on different optimization techniques. BASES uses a small-dimensional search space by modifying the weights given to each surrogate based on the feedback received from querying the target, while TREMBA learns a low-dimensional embedding then performs an efficient search within this space. As discussed in 2.2, all query attacks in this study require

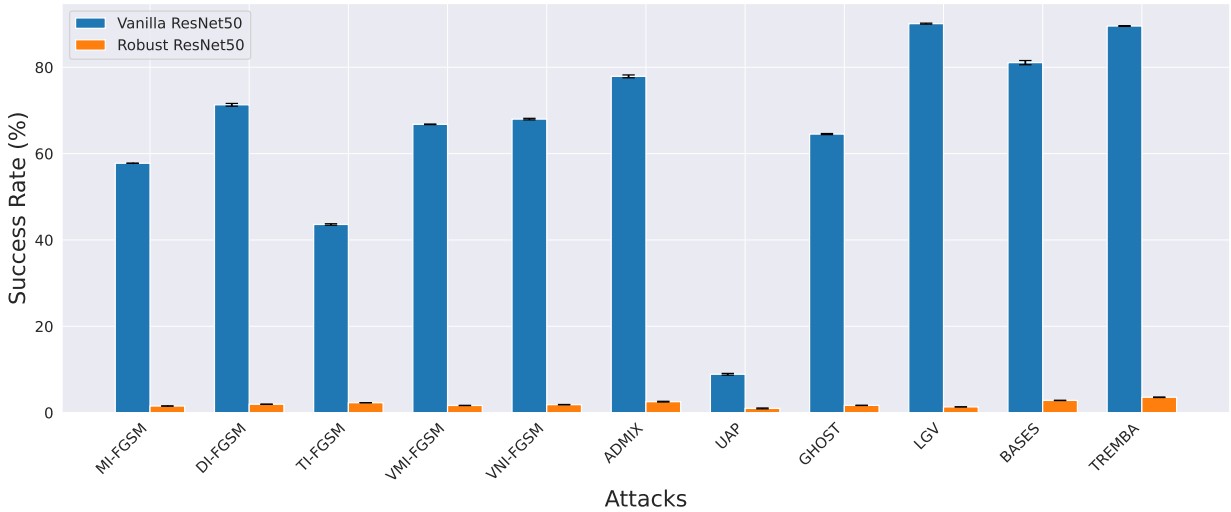

Figure 1: The blue bars (with error bars) show the success rates and standard deviations for the vanilla ResNet50 model, while the orange bars (with error bars) show the results for the robust ResNet50 model.

surrogate models. The hyperparameters for all transfer and query attacks are detailed in the Appendix B and C, respectively.

We employ surrogate models that are individually larger than the target model or collectively larger as an ensemble, allowing us to leverage the full capacity of the attack. For all single-surrogate attacks, including the five FGSM-based baseline attacks, ADMIX, UAP, GHOST, and LGV, the surrogate is a WideResNet50-2 (Zagoruyko, 2016) with standard training. For TREMBA, the surrogates are an ensemble of four models with standard training: VGG16 (Simonyan & Zisserman, 2014), ResNet18, SqueezeNet (Iandola et al., 2016), and GoogLeNet (Szegedy et al., 2015) as provided in their repository. For BASES, the surrogates are an ensemble of 10 models with standard training: VGG16$_{BN}$, ResNet18, SqueezeNet$_{11}$, GoogLeNet, MnasNet$_{10}$ (Tan et al., 2019), DenseNet161 (Huang et al., 2017), EfficientNet$_{B0}$ (Tan & Le, 2019), RegNet$_{Y400MF}$ (Radosavovic et al., 2020), ResNeXt101$_{32x8d}$ (Xie et al., 2017), ConvNeXt$_{Small}$ (Liu et al., 2022) from . The models used in this study were obtained from the torchvision library (maintainers & contributors, 2016) and the RobustBench benchmark Croce et al. (2020). We excluded the SGM attack from this section because, at the time of this study, its implementation supports only ResNet and DenseNet surrogates, while our setup uses WideResNet50-2.

### 3.2 Results

We evaluated the ASR of the black-box attacks against standard and adversarially trained Resnet-50 models. We illustrate the mean ASR using bars and the standard deviation (STD) as error bars in Figure 1. Against the undefended model, all attacks except UAP achieve more than 40% ASR, with the most recent attacks LGV, BASES and TREMBA reaching 90.11% ± 0.12, 81.08% ± 0.49, and 89.56% ± 0.09, respectively. However, all black-box attacks fail against the robustified model with an ASR of less than 4%. The most recent attacks only bring about marginal improvement over the simplest FGSM attacks. The most effective attack against the defended model in our evaluation, TREMBA achieves 3.56% ASR while TI-FGSM reaches 2.81% ASR. To ensure that the reduced effectiveness of black-box attacks against the adversarially trained ResNet50 model is not due to budget limitation, we quadruple the budget of two of the best black-box attacks in our study (BASES and TREMBA). Even with this increased budget, both attacks still significantly underperform against adversarially trained ResNet50 compared to their performance on the vanilla ResNet50 model, as detailed in the Appendix E.

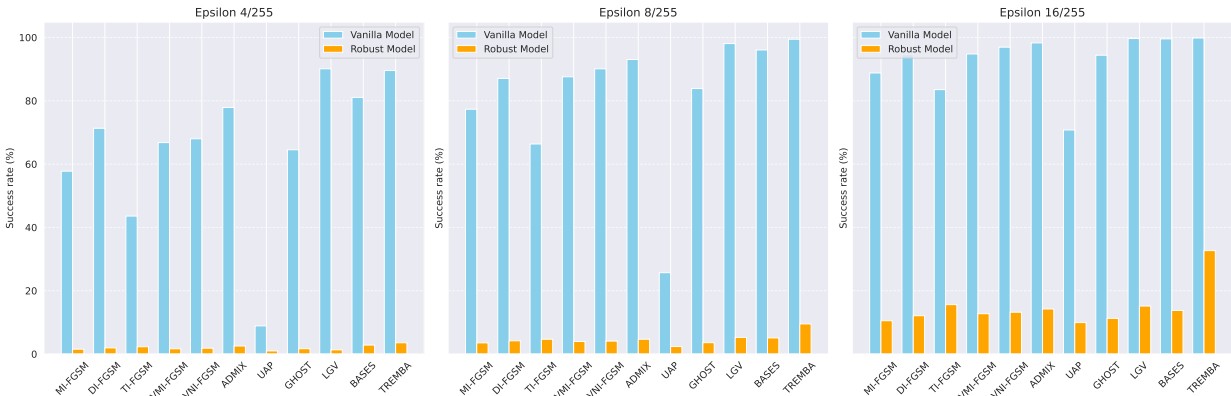

Figure 2: Ablation study of the impact of the perturbation budget $\epsilon$.

### 3.3 Ablation study

We study in the following the impact of increasing perturbation and computation budgets. In Fig.2 we present the results for the increasing $\epsilon$ budgets, and report the exact values and the ablation results on the number of iterations and queries, both using $\epsilon = 16/255$, in Appendix F. Our results from Fig.2 confirm that significantly increasing the perturbation budget does not lead to a collapse of robustness of adversarially trained models and that TREMBA is the only attack with a success rate increasing over 30% for extremely large budgets. The results in the Appendix in Table 13 show that increasing iterations has negligible impact, and Table 14 that increasing the number of queries has also negligible impact.

> **Insight 1**
>
> The effectiveness of black-box attacks drastically decreases against a simple adversarially trained model, with the strongest attack seeing its ASR drop from 89.56% to 3.56%.

## 4 Effectiveness of White-Box Defenses Against Black-Box Attacks

Given that even the most recent advances in black-box attacks remain ineffective against Madry adversarial training, we investigate whether recent innovations in defense mechanisms (studied in white-box settings) further reduce the effectiveness of black-box attacks. In particular, we want to observe whether these defenses designed against white-box attacks (viz. AutoAttack) and established benchmarks (viz. Robustbench), generalize to black-box attacks. A positive answer to these questions would maintain (and even raise) the motivation to develop defenses in white-box settings, whereas a negative answer would raise an urgent necessity of investigating specific defenses against black-box attacks.

### 4.1 Experimental Setup

We followed the same protocol as in the previous section, in terms of datasets, and adversarial black-box attacks. We selected nine robust target models from the Robustbench leaderboard. We ensured to have an even distribution across the robustness spectrum for the nine targets. The robust accuracy of the most robust model, a Swin-L (Liu et al., 2024) is 59.64% (that is, a success rate of AutoAttack of 25.97%), and the robust accuracy of the least robust model, a ResNet18 (Salman et al., 2020) is 25.32% (for a success rate of AutoAttack of 50.53%). Table 5 presents the architecture and the number of parameters of the models. These nine models have been defended using different categories of mechanisms, listed in Table 6.

We kept the same surrogate ensemble for BASES and TREMBA, and we used the surrogate model ResNet50 with standard training for single-surrogate attacks. In addition to the selected black-box attacks, we incorporated the SGM attack due to its compatibility with the ResNet50 surrogate model.

Table 1: Summary of different models from the RobustBench leaderboard (Croce et al., 2020) and their configurations. Legend of the defenses in Table 6.

| Rank | Reference | Architecture | Robust Accuracy | Defense | Architecture Type | Parameters |
|---|---|---|---|---|---|---|
| 1 | Liu et al. (2024) | Swin-L (Liu et al., 2021) | 59.56 % | A | Transformer | 196.53M |
| 2 | Bai et al. (2024) | ConvNeXtV2+Swin-L (Liu et al., 2022; 2021) | 58.50 % | A, B | Transformer & Convolution | 394.49M |
| 3 | Liu et al. (2024) | ConvNeXt-L (Liu et al., 2022) | 58.48 % | A | Convolution | 197.77M |
| 5 | Liu et al. (2024) | Swin-B (Liu et al., 2021) | 56.16 % | A | Transformer | 87.77M |
| 7 | Liu et al. (2024) | ConvNeXt-B (Liu et al., 2022) | 55.82 % | A | Convolution | 88.59M |
| 12 | Singh et al. (2024) | ViT-S+ConvStem (Dosovitskiy et al., 2020; Xiao et al., 2021) | 48.08 % | C | Transformer | 22.78M |
| 17 | Salman et al. (2020) | WideResNet50.2 (Zagoruyko, 2016) | 38.14 % | D | Convolution | 68.88M |
| 18 | Salman et al. (2020) | Resnet50 (He et al., 2016) | 34.96 % | D | Convolution | 25.56M |
| 21 | Salman et al. (2020) | Resnet18 (He et al., 2016) | 25.32 % | D | Convolution | 11.69M |

Table 2: List of the defense mechanisms.

| Label | Description |
|---|---|
| A | Adv training with large data augmentation, regularisation, weights averaging, pretraining |
| B | Non linear ensemble of two base models robust and vanilla |
| C | Downsample convolutional layers before subsequent network layers |
| D | Standard adversarial training |

## 4.2 Results

We analyze the performance of defenses against black-box attacks by examining the impact of different defense strategies, the size of the models, and the architecture of the models, as well as the relationship between AutoAttack success rates and black-box attack success rates.

**Impact of defense mechanism.** We categorize the models defended in Table 5 into four distinct families (A, B, C, and D), reflecting the various training strategies used to improve model robustness. In Figure 3 we show the ASR of the 12 black-box attacks against the four defenses (robust models). The models are sorted according to the Robustbench leaderboard, with stronger defenses against AutoAttack positioned to the right. Models using the same category of defense mechanisms but differing architectures are grouped in the same color. We observe that stronger defenses against AutoAttack lead to stronger defenses against black-box attacks. For instance, Defense A (red) outperforms Defense C (yellow), which in turn surpasses Defense D (blue) in all black-box attacks. An exception to this trend is observed with the Conv + Swin Defense ensemble, ranked # 2 on Robustbench, and the mixing mechanisms of types A and B. Although ranked second against AutoAttack, it performs similarly to Defense D when subjected to black-box attacks. This is explained by the composition of defense mechanisms A and B, which combines the adversarially trained model from Defense A with a standard model (vanilla), as explained in Section 2.3. This hybrid nature introduces a vulnerability: while this defense ranks among the best defenses against AutoAttack, the inclusion of a standard model component causes its performance against black-box attacks to align with that of weaker AutoAttack defenses. This vulnerability arises because black-box attacks often exploit vanilla surrogates, making the defense susceptible due to the presence of the standard model.

This trend remains consistent when increasing the epsilon values from 4/255 to 8/255 and 16/255, as well as when extending the iteration budget from N to 2N and 5N. Similarly, increasing the query budget from N to 2N does not alter the observed pattern. The detailed results supporting this conclusion are presented in Appendix F.

**Impact of model size.** We compare in Table 3 the success rates of the black-box attacks against similar adversarially trained models. These models do not benefit from advanced defense mechanisms (e.g., synthetic data augmentation, ...) and only differ in size. While larger models with adversarial training tend to achieve higher robustness to AutoAttack, the increase in size does not lead to lower success rates. Similar insights are revealed in Table 4, where we compare the robustness of two sizes of ConvNext models and SWIN Transformers. Larger models do not consistently demonstrate stronger robustness to adversarial perturbations in black-box settings, although increased size improves robustness against AutoAttack.

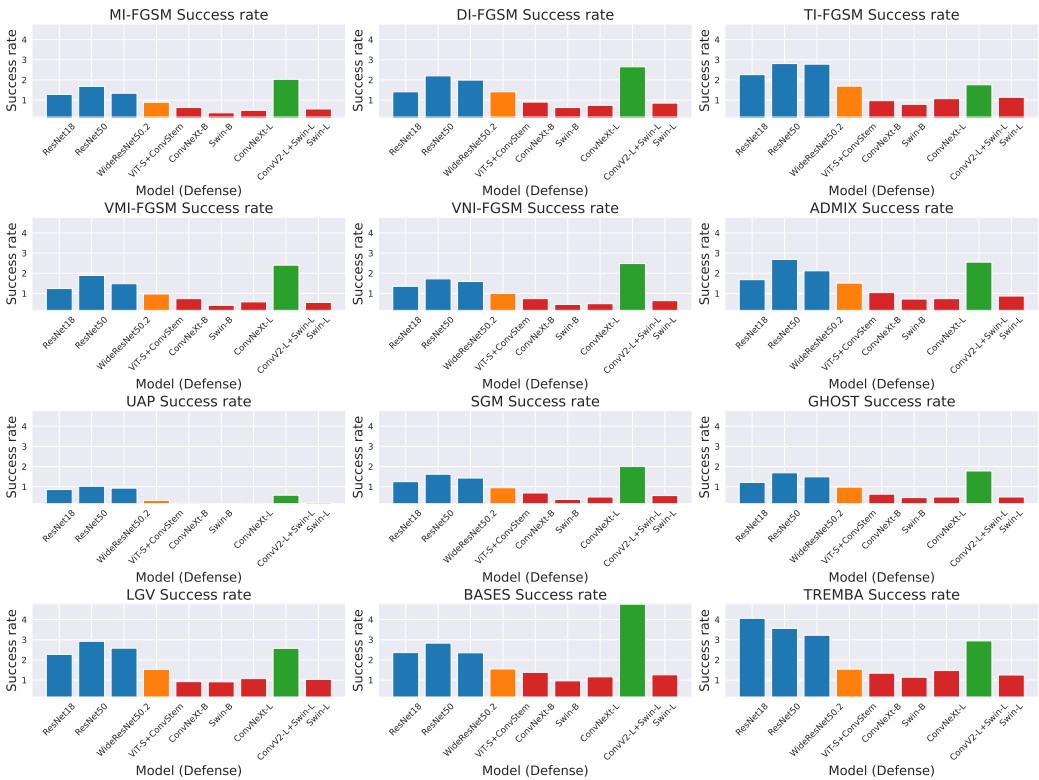

Figure 3: Success rate of blackbox attacks against SoTA defenses.

Table 3: Success rate on Resnet models trained by Salman et al. (2020).

| Attacks | Salman ResNet18 | Salman ResNet50 | Salman WideResNet50.2 |
|---|---|---|---|
| **AUTOATTACK** | 50.53 | 43.84 | 42.87 |
| **MI-FGSM** | $1.28 \pm 0.00$ | $1.68 \pm 0.00$ | $1.34 \pm 0.00$ |
| **DI-FGSM** | $1.41 \pm 0.02$ | $2.20 \pm 0.10$ | $1.99 \pm 0.04$ |
| **TI-FGSM** | $2.26 \pm 0.04$ | $2.81 \pm 0.07$ | $2.78 \pm 0.10$ |
| **VMI-FGSM** | $1.24 \pm 0.05$ | $1.89 \pm 0.04$ | $1.48 \pm 0.05$ |
| **VNI-FGSM** | $1.35 \pm 0.02$ | $1.72 \pm 0.03$ | $1.59 \pm 0.05$ |
| **ADMIX** | $1.68 \pm 0.06$ | $2.68 \pm 0.13$ | $2.12 \pm 0.08$ |
| **UAP** | $0.86 \pm 0.10$ | $1.02 \pm 0.01$ | $0.93 \pm 0.07$ |
| **SGM** | $1.25 \pm 0.00$ | $1.62 \pm 0.00$ | $1.43 \pm 0.00$ |
| **GHOST** | $1.21 \pm 0.18$ | $1.69 \pm 0.05$ | $1.49 \pm 0.06$ |
| **LGV** | $2.27 \pm 0.11$ | $2.92 \pm 0.01$ | $2.58 \pm 0.09$ |
| **BASES** | $2.36 \pm 0.03$ | $2.83 \pm 0.04$ | $2.35 \pm 0.06$ |
| **TREMBA** | $4.06 \pm 0.06$ | $3.56 \pm 0.06$ | $3.22 \pm 0.06$ |

**Impact of model architecture.** Liu et al. (2024) demonstrate that the best robustness against AutoAttack can be achieved using indiscriminately Convnets or Transformers models. Our results in Table 4 confirm that for a fixed size, both architectures demonstrated similar robustness against black-box attacks.

**Relationship between AutoAttack success rate and black-box success rate.** As shown in Figure 4, the scatter plot illustrates the relationship between the ASR of AutoAttack and that of 12 black-box attacks. Each data point represents the ASR on the y-axis of a given black-box attack (indicated by a color) against a specific robust model (determined by its AutoAttack success rate on the x-axis).

Table 4: Success rate on very large models trained by Liu et al. (2024).

| Attacks | Liu ConvNeXt-B | Liu Swin-B | Liu ConvNeXt-L | Liu Swin-L |
|---------|----------------|------------|----------------|------------|
| AUTOATTACK | 28.01 | 27.14 | 25.95 | 25.98 |
| MI-FGSM | $0.63 \pm 0.00$ | $0.37 \pm 0.00$ | $0.49 \pm 0.00$ | $0.56 \pm 0.00$ |
| DI-FGSM | $0.90 \pm 0.02$ | $0.63 \pm 0.03$ | $0.74 \pm 0.09$ | $0.85 \pm 0.04$ |
| TI-FGSM | $0.97 \pm 0.02$ | $0.79 \pm 0.07$ | $1.07 \pm 0.03$ | $1.14 \pm 0.08$ |
| VMI-FGSM | $0.74 \pm 0.02$ | $0.41 \pm 0.01$ | $0.58 \pm 0.03$ | $0.55 \pm 0.01$ |
| VNI-FGSM | $0.74 \pm 0.04$ | $0.45 \pm 0.02$ | $0.49 \pm 0.02$ | $0.64 \pm 0.04$ |
| ADMIX | $1.04 \pm 0.04$ | $0.72 \pm 0.02$ | $0.74 \pm 0.01$ | $0.87 \pm 0.00$ |
| UAP | $0.19 \pm 0.04$ | $0.16 \pm 0.02$ | $0.18 \pm 0.06$ | $0.21 \pm 0.04$ |
| SGM | $0.69 \pm 0.00$ | $0.37 \pm 0.00$ | $0.49 \pm 0.00$ | $0.56 \pm 0.00$ |
| GHOST | $0.63 \pm 0.03$ | $0.46 \pm 0.07$ | $0.49 \pm 0.09$ | $0.49 \pm 0.04$ |
| LGV | $0.92 \pm 0.07$ | $0.91 \pm 0.03$ | $1.07 \pm 0.03$ | $1.03 \pm 0.03$ |
| BASES | $1.38 \pm 0.06$ | $0.96 \pm 0.02$ | $1.16 \pm 0.00$ | $1.26 \pm 0.02$ |
| TREMBA | $1.34 \pm 0.04$ | $1.14 \pm 0.03$ | $1.47 \pm 0.01$ | $1.25 \pm 0.01$ |

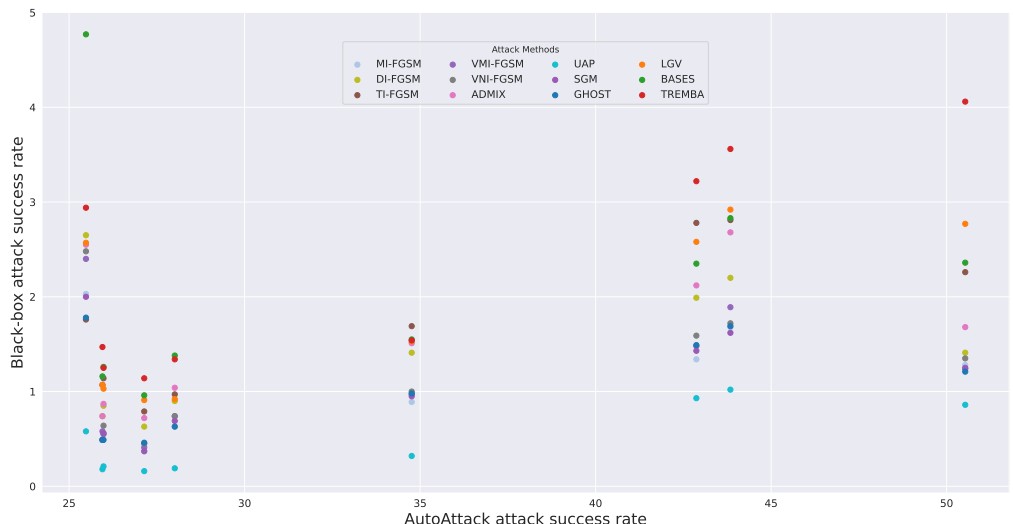

Figure 4: Relation between success rate of AutoAttack and the success rate of black-box attacks.

Although the initial ASR of black-box attacks is low (less than 5%), we observe a clear downward trend as robust models improve their effectiveness against AutoAttack, until the best four robust models (with AutoAttack success rate ranging from 25.48% to 27.14%). This vulnerability stems from the lack of correlation between model architecture and robustness, as these models all employ defense (A) with diverse sizes and architectures, as shown in Table 5.

Furthermore, the best robust model (25.48% AutoAttack success rate) underperforms compared to the other top robust models against black-box attacks. This occurs because it uses the ensemble defense (B), which, as discussed earlier, combines a robust model with a standard model, making it vulnerable to black-box attacks that often exploit surrogates resembling standard models.

> **Insight 2**
>
> SoTA defenses against AutoAttack generally correlate with improved black-box robustness. However, having a large model size does not always guarantee better robustness. Additionally, an ensemble defense that achieves similar robustness to a single defense against AutoAttack may be less effective against black-box attacks, especially those using surrogate models similar to the ensemble's components.

## 5 Robust surrogates to Improve Black-Box Attacks

The results from the previous section demonstrated that models robust against AutoAttack remain relevant (to some extent) against black-box attacks. A critical concern then is whether these robust "defenses" could not be used by the attacker to improve its black-box attack. Indeed, most of the black-box require designing a surrogate and can therefore leverage the recent advances in model defense to strengthen their attacks. We hypothesize that these new defenses raise security risks by providing stronger surrogates to attackers.

### 5.1 Experimental Setup

We adhered to the same protocol as in previous sections, maintaining consistency in datasets and adversarial black-box attacks. We considered various configurations regarding the robustness of both the surrogate and target models. For the target models, we selected highly robust models (Swin-L, ConvNeXtV2+Swin-L, ConvNext-L from (Liu et al., 2024) and (Bai et al., 2024)), a moderately robust model (ResNet18 from (Salman et al., 2020)), and a non-robust model (ResNet50). This way, we can measure the effects of using robust surrogates on targets with top defense (A), a simple defense (D) and no defense.

As for surrogate, we used robust and non-robust models. The selection of robust surrogates was based on choosing the highly ranked robust models that are not used as target and are compatible with the attack methods. Hence, we used the robust surrogate RaWideResNet-101-2 from (Peng et al., 2023) for single surrogate attacks, because it is the best defense in Robustbench that supports skip connections (which are essential for the GHOST attack). For ensemble surrogate attacks, we also added the robust surrogate Swin-B from (Liu et al., 2024), as it is the best robust model available, excluding the target models. For single, non-robust surrogate attacks, we used a vanilla WideResNet-101-2 in order to have the same architecture as in the robust surrogate, ensuring fair comparison. For BASES and TREMBA (who use an ensemble of surrogates), we kept the same ensemble of vanilla models as in our previous experiments. We excluded the SGM attack that does not support the WideResNet-101-2 surrogates used in our setup.

### 5.2 Results

We present in Figure 5 the success rate of black-box attacks using non-robust surrogates and robust surrogates against a non robust target and robust targets.

**Non-robust target:** The results clearly show that using robust surrogate models is detrimental to attack success rates. Across all black-box attacks, non-robust surrogates consistently outperform robust surrogates, with an average improvement of 35.19 percentage points in attack success rates on the vanilla ResNet50 model. For instance, GHOST improves from 17.95% to 49.36%, BASES from 17.94% to 81.08%, and TREMBA from 67.54% to 89.56%. Even UAP that had the lowest success rate improves from 0.73% to 7.11%. Notably, attacks that leveraged a training phase before the attacking phase (LGV and TREMBA) exhibited a relatively smaller improvement gap compared to other attacks as robust surrogates were already performing well. Specifically, LGV improved from 81.34% to 87.39%, while TREMBA increased from 67.54% to 89.56%.

**Robust target:** By contrast, robust surrogates yield significant improvement in the success rate of black-box attacks when targeting robust models, resulting in an average improvement of 6.42 percentage points in ASR across attacks and target models. Notably, attacks using robust surrogates on top-tier robust models (e.g., Liu ConvNeXt-L and Liu Swin-L) yield up to 15 times higher success rates than their non-robust

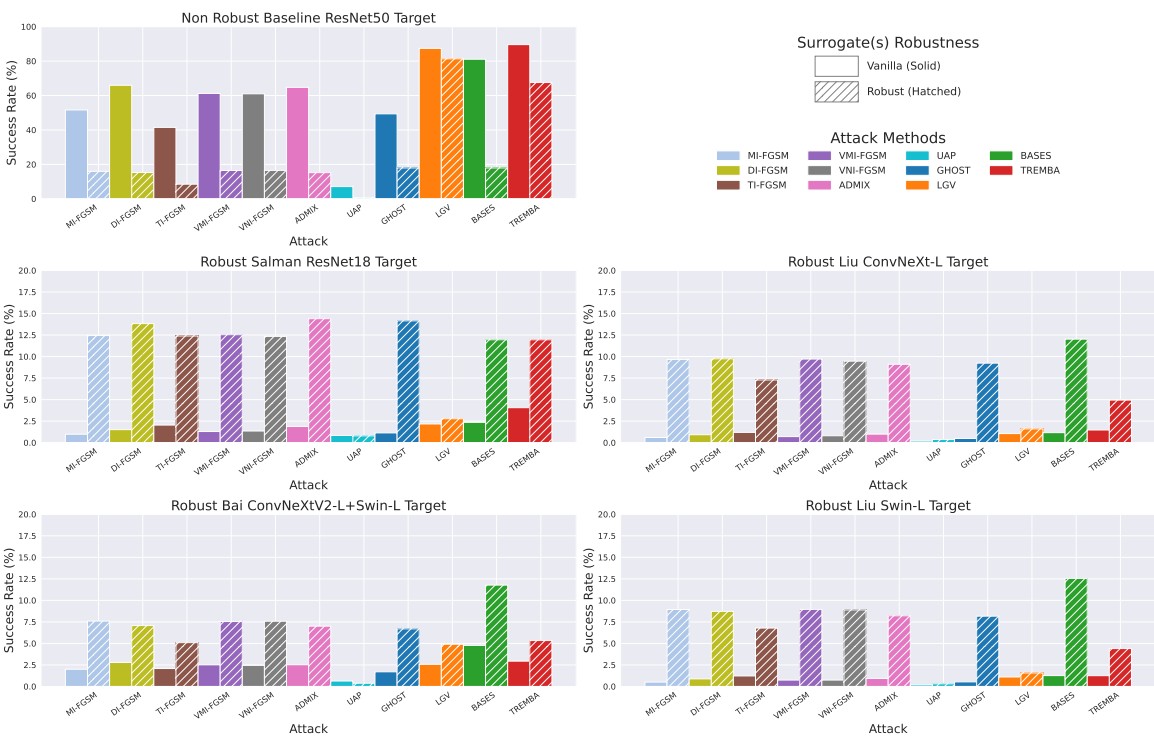

Figure 5: Success rates of black-box attacks using vanilla and robust surrogates against a vanilla target and robust targets.

counterparts. For instance, when using the GHOST attack to target the highly robust Swin-L model, the robust surrogate achieves a success rate of 8.16%, compared to just 0.53% with the non-robust surrogate. Similarly, BASES improves its success rate from 1.26% to 12.55% against the Swin-L model. An improvement of greater than or equal to 2 percentage points in ASR is consistent across all defense mechanisms and all attacks, except LGV and UAP. We hypothesize that the initial training phase of the LGV models, where a high learning rate is used over additional epochs, potentially weakens the model's robustness during the early sampling phase, which contrasts with GHOST networks that maintain stronger robustness after their creation. This hypothesis is explored in Appendix G.

The results highlight that the choice of surrogate model plays a critical role in the effectiveness of transfer-based black-box attacks, particularly in restricted environments where the robustness of the target is unknown. Robust models, by design, tend to secure blind spot regions in the input space in which vanilla models often fail to predict adversarial examples correctly. As a result, transfer attacks on robust surrogates aim to fool other regions of the input space that are not yet secured by adversarial training, making them better suited for attacking robust models. On the other hand, transfer attacks on vanilla surrogates focuses more on blind spot regions of the input space that are also likely to be mispredicted by other vanilla target models.

> **Insight 3**
>
> The choice of the surrogate model, robust or vanilla, impacts the success of black-box attacks. Robust surrogates are more effective against robust targets, while vanilla surrogates excel against vanilla targets. Attackers can therefore improve attack effectiveness by adaptively selecting surrogates based on the perceived robustness of the target model.

## 6 Limitations and Perspectives

While our work investigated a large set of black-box attacks and robust models, we mostly focused on extremely robust models. Zhang et al. (2024) for instance focused on mildly perturbed examples and uncovered their interactions with simple adversarial training. Expanding our study of transfer-based attacks to consider varying levels of attack strength and robustness levels between surrogate and target models would also help to deepen our understanding of transferability dynamics. In particular, considering randomness in the pre-processing, and defenses including obfuscation and randomness.

In addition, while this study focused on image classification tasks, future research could extend the analysis to other tasks, such as object detection or segmentation, to understand how black-box robustness may vary across applications. These regression tasks raise a new challenge of defining acceptable robustness thresholds, given that the output is not binary. Although there is an established benchmark for adversarial robustness for classification tasks, there is no such benchmark for segmentation tasks.

Another limitation is that while we included large transformers models, we did not explore the robustness of foundations models such as GPT Radford et al. (2018), LLaMA Touvron et al. (2023), CLIP Radford et al. (2021), and SAM Kirillov et al. (2023), or their variants. The research is still active to achieve effective and aligned foundation models, exploring robustification mechanisms for these models is still in its infancy.

## 7 Conclusion

This research investigated the effectiveness of black-box adversarial attacks against SoTA and standard robust models using ImageNet, structured around three series of experiments, uncovering key insights.. Our first findings demonstrate that even advanced black-box attacks often struggle against a relatively simple adversarial defense. This suggests that adversarial training can be a highly effective baseline defense for real-world systems accessible via APIs. Consequently, understanding the limitations of current black-box attacks against robust models is crucial for developing more effective attack strategies. Based on our second experiment, we found that defense mechanisms optimized against AutoAttack often generalize effectively to resist black-box attacks, even though performance differences are less pronounced in black-box settings.

This finding supports the relevance of AutoAttack as a benchmark for black-box robustness and suggests that current efforts focused on white-box defenses provide a meaningful degree of robustness in black-box scenarios as well – though ensemble defenses may not always translate their AutoAttack robustness to black-box robustness, especially against attacks using similar surrogate models as the ensemble. Finally, we observed that selecting a surrogate with a level of robustness aligned with that of the target can significantly impact the success rate of transfer-based attacks. This finding reveals the possibility for strategic selection of surrogate models when planning black-box attacks. Future attacks should consider designing black-box attacks that specifically target SOTA robust models to better reflect real-world defensive capabilities. Additionally, further exploration of why SoTA defenses optimized for AutoAttack are also relevant for black-box robustness could yield valuable insights into cross-attack resilience mechanisms. In conclusion, our study demonstrates that SoTA defenses are notably resilient against black-box attacks, underscoring the importance of developing more targeted attack strategies to effectively challenge these modern robust models. As the field advances, these insights can guide both the creation of more sophisticated black-box attacks and the design of improved defense mechanisms that support the deployment of robust AI systems across a wide range of applications.

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

## Appendix

## A   Models

Table 5: Summary of different models from Robustbench Croce et al. (2020) and their configurations. Legend of the defenses in Table 6

| Rank | Reference | Model Name | Robust Accuracy | Defense | Architecture | Parameters |
|---|---|---|---|---|---|---|
| 1 | Liu et al. (2024) | Swin-L | 59.56 % | A | Transformer | 196.53M |
| 2 | Bai et al. (2024) | ConvNeXtV2+Swin-L | 58.50 % | A, B | Transformer & Convolution | 394.49M |
| 3 | Liu et al. (2024) | ConvNeXt-L | 58.48 % | A | Convolution | 197.77M |
| 5 | Liu et al. (2024) | Swin-B | 56.16 % | A | Transformer | 87.77M |
| 7 | Liu et al. (2024) | ConvNeXt-B | 55.82 % | A | Convolution | 88.59M |
| 12 | Singh et al. (2024) | ViT-S+ConvStem | 48.08 % | C | Transformer | 22.78M |
| 17 | Salman et al. (2020) | WideResNet50.2 | 38.14 % | D | Convolution | 68.88M |
| 18 | Salman et al. (2020) | Resnet50 | 34.96 % | D | Convolution | 25.56M |
| 21 | Salman et al. (2020) | Resnet18 | 25.32 % | D | Convolution | 11.69M |

Table 6: List of the defense mechanism.

| Label | Description |
|---|---|
| A | Adv training with large data augmentation, regularisation, weights averaging, pretraining |
| B | Non linear ensemble of two base models robust and vanilla |
| C | Downsample convolutional layers before subsequent network layers |
| D | Standard adversarial training |

## B  Hyperparameters of Transfer Attacks

Tables 7 and 8 provide a comprehensive overview of the hyperparameters used for the ten transfer-based adversarial attacks in our experiments. They categorize the key hyperparameters into two groups: common hyperparameters shared across most attacks and attack-specific hyperparameters that define the unique characteristics of each method. Most of the attack hyperparameters were kept at their default values, as specified in the repositories or libraries that provide the implementations of adversarial attacks (Kim, 2020; Woo, 2024).

### B.1  Common Hyperparameters

Table 7 provides the default values and explanations for these common hyperparameters, such as the step size (Alpha), momentum factor (Decay), and number of iterations (Steps), which are essential for iterative attacks as they control the perturbation magnitude and the optimization process.

### B.2  Attack-Specific Hyperparameters

In addition to the shared hyperparameters, each attack incorporates its own unique hyperparameters tailored to its specific mechanisms. Table 8 summarizes these attack-specific hyperparameters, listing the default values, explanations, and the corresponding attacks to which they apply.

## C  Hyperparameters of Query Attacks

Tables 9 and 10 provides an overview of the hyperparameters used in BASES and TREMBA, respectively. Most of these hyperparameters were kept at their default values, as specified in their repositories, since our experiments aimed to evaluate their performance under standard configurations. For BASES, key hyperparameters include the ensemble size (N_wb), the type of fusion for surrogate models (Fuse), and the number of weight updates (Iterw). Similarly, TREMBA, which operates in two phases (training and attacking), relies on hyperparameters such as the number of training epochs (Epochs), the number of attack iterations (Num_Iters), the number of samples for Natural Evolution Strategy (NES), and latent space parameters like the Gaussian noise standard deviation (Sigma).

## D  Computational Budget of Attacks

Table 11 details the computational budgets of all studied black-box attacks, quantified as the number of model calls required during the attacking phase per entry.

## E  Increasing the Computation Budget of Black-Box Attacks

We are keeping the perturbation budget $\epsilon = 4/255$, and quadrupled (multiplied by 4) the budget of the best black-box attacks in section 3 (BASES and TREMBA) against the robust ResNet50 model, to reassure that black-box attacks reduced effectiveness (compared to their effectiveness against the vanilla ResNet50 model) is not due to a limited budget. For BASES, we increased the number of inner iterations of PGD

Table 7: Common hyperparameters for transfer-based attacks.

| PARAMETER | DEFAULT VALUE | EXPLANATION | APPLIES TO |
|---|---|---|---|
| Alpha | 2/255 | Step size or update rate for perturbation. | All attacks except UAP |
| Decay | 1.0 | Momentum factor. | All attacks except UAP |
| Steps | 10 | Number of iterations. | All attacks except UAP |
| Epsilon (eps) | 4/255 | Maximum perturbation for adversarial example. | All attacks |

Table 8: Attack-specific hyperparameters for transfer-based attacks.

| PARAMETER | DEFAULT VALUE | EXPLANATION | APPLIES TO |
|---|---|---|---|
| Resize Rate | 0.9 | Rate at which the input images are resized. | DIFGSM, TIFGSM |
| Diversity Prob | 0.5 | Probability to apply input diversity. | DIFGSM, TIFGSM |
| Kernel Name | Gaussian | Type of kernel used. | TIFGSM |
| Kernel Length | 15 | Length of the kernel. | TIFGSM |
| Sigma (nsig) | 3 | Radius of the Gaussian kernel. | TIFGSM |
| N | 5 | Number of samples in the neighborhood. | VMI, VNI |
| Beta | 1.5 | Upper bound for neighborhood. | VMI, VNI |
| Prior Type | no_data | Type of prior used for attack optimization. | UAP |
| Patience Interval | 5 | Number of iterations to wait before checking convergence. | UAP |
| Gamma | 0.5 | Decay factor for gradients from skip connections. | SGM |
| LGV Epochs | 5 | Number of epochs for training the LGV models. | LGV |
| LGV Models Epoch | 8 | Number of models collected per epoch. | LGV |
| LGV Learning Rate | 0.1 | Learning rate for the LGV training phase. | LGV |
| LGV Batch Size | 256 | Batch size for the LGV training phase. | LGV |
| Randomized Modulating Scalar | 0.22 | Drawn from the uniform distribution 1-scalar, 1+scalar. | GHOST |
| Portion | 0.2 | Portion for the mixed image. | Admix |
| Size | 3 | Number of randomly sampled images. | Admix |

Table 9: Hyperparameters for the BASES attack.

| PARAMETER | DEFAULT VALUE | EXPLANATION |
|---|---|---|
| Epsilon (eps) | 4 | Perturbation bound magnitude (0 to 255 pixel range)). |
| N_wb | 10 | Number of models in the ensemble. |
| Bound | linf | Perturbation bound norm type. |
| Iters | 10 | Number of inner iterations. |
| Fuse | loss | Fuse method, e.g., loss or logit. |
| Loss_name | CW | Loss function used for optimization. |
| Algo | PGD | White-box algorithm used for the perturbation machine |
| X | 3 | Factor to scale the step size alpha. |
| Learning Rate (lr) | 0.005 | Learning rate for weight updates. |
| Iterw | 20 | Number of iterations to update weights. |

within the perturbation machine (Iters) to 40. For TREMBA, we increased the Number of iterations for NES (Num_Iters) to 800.

Figure 6 presents the success rates of BASES and TREMBA attacks against the vanilla ResNet-50 model with standard attack budget, and against the adversarially trained ResNet-50 model under both standard and quadrupled attack budgets. Both attacks underperform against the adversarially trained ResNet50 under quadrupled budgets compared to the vanilla model with standard budgets. For instance, BASES achieves 2.56% ASR on the robust ResNet50 Budget (x4) versus 81.08% on the vanilla ResNet50. Similarly, TREMBA's ASR is at 5.24% on the robust ResNet50 Budget (x4) compared to 89.56% on the vanilla ResNet50. These findings confirm that our earlier results in section 3 were not a consequence of budget limitations.

## F   Increasing the Perturbation Budget of Black-Box Attacks

We have increased the perturbation budget to cover $\epsilon \in \{4/255, 8/255, 16/255\}$. In Table 12, Table 15, and Table 17, we only change the perturbation budget $\epsilon$.

Table 10: Hyperparameters for the TREMBA attack.

| PARAMETER | DEFAULT VALUE | EXPLANATION |
|---|---|---|
| Epsilon (eps) | 8/255 (Training Phase), 4/255 (Attacking Phase) | Perturbation bound magnitude. |
| Learning Rate (G) | 0.01 (Training Phase), 5.0 (Attacking Phase) | for gradient updates, for latent updates |
| Momentum | 0.9 (Training Phase), 0.0 (Attacking Phase) | for SGD, for attack on the embedding space. |
| Num_images | 49k (Training Phase) | Size of the training set |
| Epochs | 500 | Number of epochs for Training. |
| Schedule | 10 | Epochs between Training learning rate decay. |
| Gamma | 0.5 | Training Learning rate decay factor. |
| Margin | 200.0 (Training Phase), 5.0 (Attacking Phase) | Margin for the loss function. |
| Sample_Size | 20 | Number of samples for NES |
| Num_Iters | 200 | Number of iterations for NES. |
| Sigma | 1.0 | Standard deviation of the Gaussian noise for NES. |
| Learning Rate Min | 0.1 | Minimum learning rate in the Attacking Phase. |
| Learning Rate Decay | 2.0 | Learning rate decay factor in the Attacking Phase. |
| Plateau_Length | 20 | Number of iterations to monitor the objective function |
| Plateau_Overhead | 0.3 | Allowed loss threshold to change before the decay |

Table 11: Computational budgets of black-box attacks

| Attack Algorithm | Number of Model Calls | Explanation |
|---|---|---|
| MIFGSM | Iterations (e.g., 10) | MIFGSM iteratively refines the adversarial image over a number of steps, each requiring a model call |
| DIFGSM | Iterations (e.g., 10) | DIFGSM performs iterative updates with input diversity applied at each step, requiring a model call per step |
| TIFGSM | Iterations (e.g., 10) | TIFGSM performs iterative updates with translation-invariant gradient smoothing at each step, requiring a model call per step |
| VMIFGSM | Iterations * (N + 1) (e.g., 60) | VMIFGSM iterates Steps times. In each iteration, there's one model call for the adversarial image and N model calls for the neighboring images to calculate the gradient variance |
| VNIFGSM | Iterations * (N + 1) (e.g., 60) | VNIFGSM, similar to VMIFGSM. In each iteration, there is one model call for the Nesterov and N model calls for the neighboring images to calculate the gradient variance |
| GHOST | Iterations (e.g., 10) | GHOST applies Gaussian noise to the skip connection during each iteration (one Ghost model per iteration), requiring one model call per step |
| LGV | Iterations (e.g., 10) | LGV uses an ensemble of models (lgv_nb_models_epoch * lgv_epochs models), with one model call at each step for each model in the ensemble |
| Admix | Iterations * Scale * Size (e.g., 150) | In each step, Admix generates Size admixed images, scales them Scale times, and then calculates the loss |
| UAP | Max Iterations + Evaluation Calls (preparing the perturbation), then 1 | The primary loop runs for max_iter iterations. Additionally, a function is called periodically to evaluate the current perturbation |
| TREMBA | (num_iters + 1) * sample_size (e.g., 4000) | Tremba iteratively refines the adversarial perturbation within the latent space of a generator network. Each iteration involves generating sample_size perturbations and evaluating their losses |
| BASES | iterw * n_iters * n_wb (e.g., 2,000) | BASES uses a surrogate ensemble containing n_wb models and performs an attack for n_iters iterations during each query (iterw) |

Once we have increased the perturbation budget to $\epsilon = 16/255$, we first study the impact of increasing the iteration budget in Table 13, and Table 16, then, we study the impact of increasing the query budget in Table 14, and Table 18.

To further examine the impact of different defense strategies with increasing epsilon values and budgets, we present the results in the form of figures. Specifically, Figure 7 and Figure 8 correspond to epsilon values of

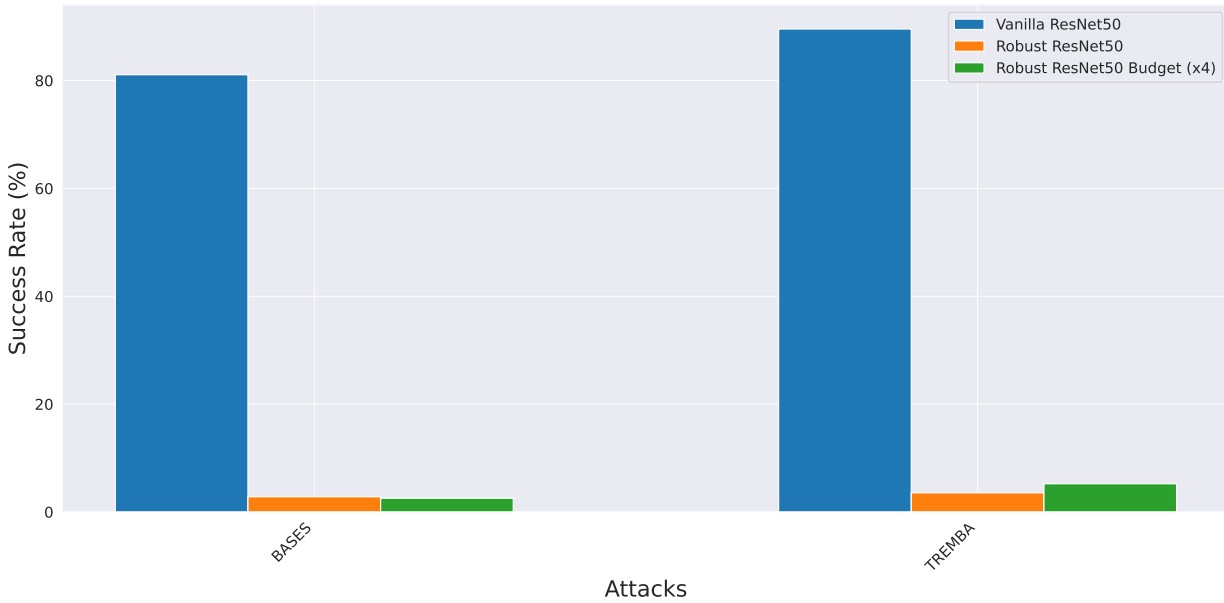

Figure 6: The blue bars show the success rate for the vanilla ResNet50 model, while the orange bars show the results for the robust ResNet50 model, and the green bars show the success rate for the ResNet50 model by quadrupling the budget of the attacks.

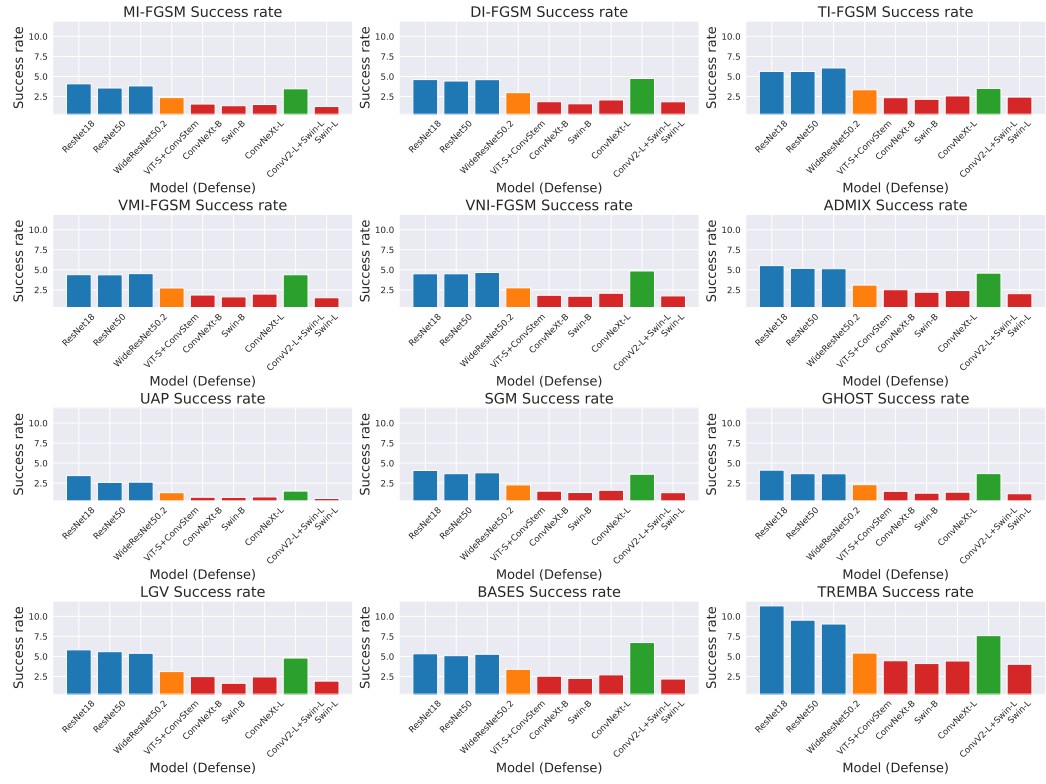

Figure 7: Success rate of blackbox attacks against SoTA defenses for epsillon 8/255.

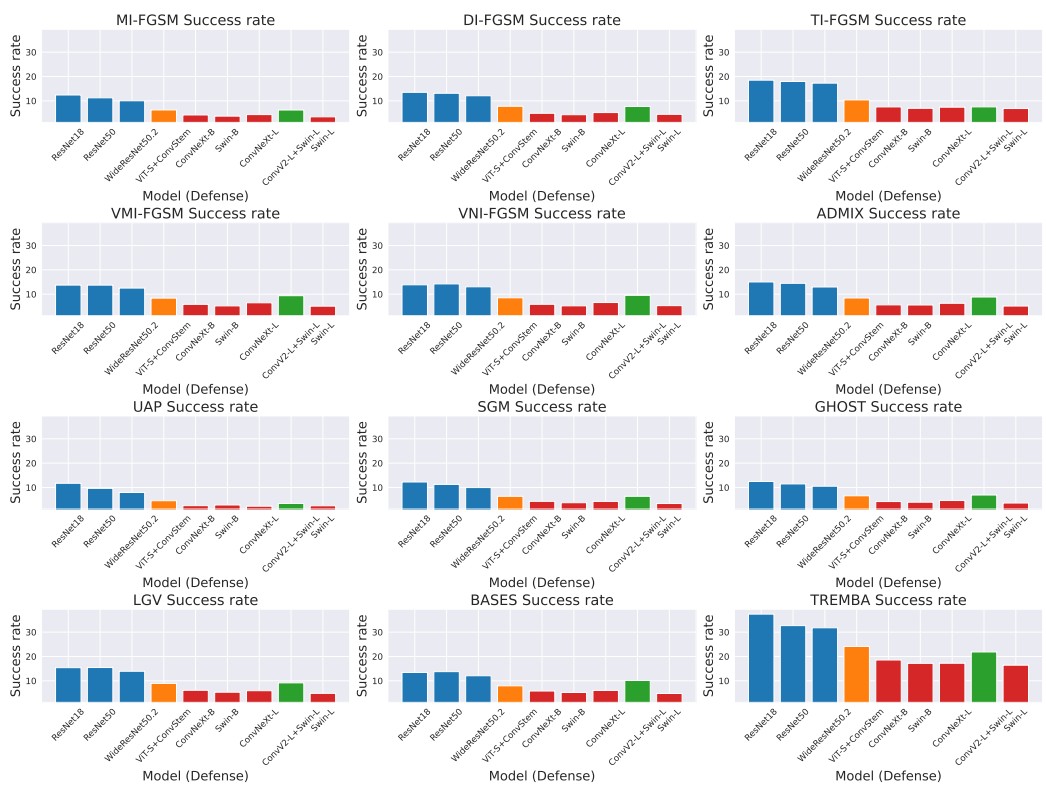

Figure 8: Success rate of blackbox attacks against SoTA defenses for epsilon 16/255.

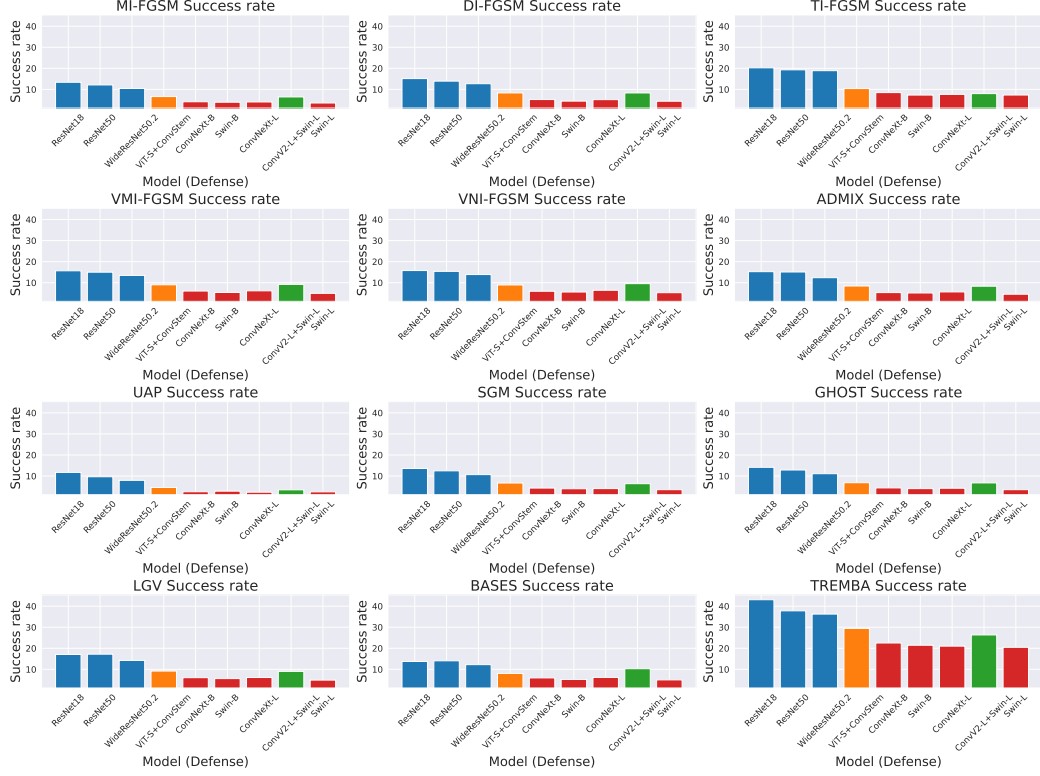

Figure 9: Success rate of blackbox attacks against SoTA defenses for epsilon 16/255 and iteration=2N.

Table 12: Attack Performance at Different Epsilon Values.

| Attack | Epsilon 4/255 | Epsilon 8/255 | Epsilon 16/255 |
|---|---|---|---|
| MI-FGSM | 57.77 → 1.53 | 77.33 → 3.53 | 88.79 → 10.52 |
| DI-FGSM | 71.31 → 1.94 | 87.07 → 4.18 | 94.55 → 12.11 |
| TI-FGSM | 43.59 → 2.30 | 66.35 → 4.68 | 83.53 → 15.61 |
| VMI-FGSM | 66.79 → 1.66 | 87.59 → 3.97 | 94.79 → 12.74 |
| VNI-FGSM | 68.00 → 1.85 | 90.10 → 4.09 | 96.94 → 13.21 |
| ADMIX | 77.90 → 2.54 | 93.04 → 4.68 | 98.28 → 14.24 |
| UAP | 8.89 → 0.98 | 25.70 → 2.37 | 70.78 → 9.93 |
| GHOST | 64.53 → 1.68 | 83.87 → 3.59 | 94.40 → 11.27 |
| LGV | 90.11 → 1.33 | 98.07 → 5.25 | 99.66 → 15.16 |
| BASES | 81.08 → 2.83 | 96.06 → 5.09 | 99.56 → 13.76 |
| TREMBA | 89.56 → 3.56 | 99.45 → 9.52 | 99.84 → 32.67 |

Table 13: Attack Performance Across Different Iteration Counts.

| Attack | Iteration=N | Iteration=2N | Iteration=5N |
|---|---|---|---|
| MI-FGSM | 88.79 → 10.52 | 91.63 → 11.61 | 93.79 → 12.67 |
| DI-FGSM | 94.55 → 12.11 | 97.32 → 13.30 | 99.17 → 14.38 |
| TI-FGSM | 83.53 → 15.61 | 89.31 → 17.61 | 93.45 → 18.69 |
| VMI-FGSM | 94.79 → 12.74 | 97.73 → 14.30 | 99.06 → 14.82 |
| VNI-FGSM | 96.94 → 13.21 | 98.62 → 14.67 | 99.45 → 14.98 |
| ADMIX | 98.28 → 14.24 | 98.90 → 14.58 | 99.43 → 15.02 |
| UAP | 70.87 → 9.93 | 70.78 → 9.93 | 72.38 → 10.02 |
| GHOST | 94.40 → 11.27 | 96.48 → 12.21 | 98.41 → 13.39 |
| LGV | 99.66 → 15.16 | 99.92 → 16.61 | 99.97 → 16.39 |

Table 14: Attack Performance Across Different Query Counts.

| Model | Attack | Query=N | Query=2N |
|---|---|---|---|
| Vanilla | BASES | 99.56 | 99.66 |
| Vanilla | TREMBA | 99.84 | 99.90 |
| Robust | BASES | 13.76 | 14.05 |
| Robust | TREMBA | 32.67 | 37.79 |

8/255 and 16/255, respectively. Additionally, Figure 9 for iteration and query budget of 2N and Figure 10 illustrate results for iteration budgets of 5N.

## G   AutoAttack Robustness of LGV and GHOST Models

To evaluate the impact of the LGV weight collection phase, and the GHOST perturbing phase on model robustness, we collected 10 models after finishing the first phase for both LGV and GHOST, prior to proceeding to the attacking phase. Specifically, the LGV variations were obtained by training the surrogate model for additional epochs using a high learning rate, during which the weights of the models were collected periodically. For GHOST networks, the variations were obtained by introducing a random perturbation to the scaling factor in the skip connections. Using the robust surrogate model *WideResNet-101-2* introduced in (Peng et al., 2023) and described in Section 5, we evaluated the robustness of the resulting LGV models and compared it with the robustness of GHOST models. The robust accuracies for the collected models are reported in Table 19.

Given that both LGV and GHOST models are derived from the same robust surrogate model, we observed a difference in their robustness. The robust accuracies for all LGV models is 0% (indicating 100% attack success

Table 15: Transfer Attack Performance Across Increased Epsilon 4/255, 8/255 and 16/255.

| Model | MI-FGSM | DI-FGSM | TI-FGSM | VMI-FGSM | VNI-FGSM | ADMIX | UAP | SGM | GHOST | LGV |
|---|---|---|---|---|---|---|---|---|---|---|
| ResNet18 | 1.28 → 4.08 → 12.4 | 1.41 → 4.61 → 13.5 | 2.26 → 5.63 → 18.49 | 1.24 → 4.39 → 13.72 | 1.35 → 4.5 → 13.84 | 1.68 → 5.52 → 15.01 | 0.86 → 3.44 → 11.72 | 1.25 → 4.08 → 12.25 | 1.21 → 4.12 → 12.48 | 2.27 → 5.82 → 15.39 |
| ResNet50 | 1.68 → 3.56 → 11.24 | 2.2 → 4.43 → 13.11 | 2.81 → 5.62 → 17.95 | 1.89 → 4.37 → 13.71 | 1.72 → 4.5 → 14.21 | 2.68 → 5.18 → 14.46 | 1.02 → 2.59 → 9.65 | 1.62 → 3.68 → 11.27 | 1.69 → 3.68 → 11.46 | 2.92 → 5.59 → 15.45 |
| WideResNet50.2 | 1.34 → 3.82 → 10.02 | 1.99 → 4.6 → 12.09 | 2.78 → 6.06 → 17.27 | 1.48 → 4.52 → 12.5 | 1.59 → 4.66 → 13.02 | 2.12 → 5.13 → 12.94 | 0.93 → 2.62 → 7.95 | 1.43 → 3.79 → 9.96 | 1.49 → 3.67 → 10.52 | 2.58 → 5.39 → 13.92 |
| ViT-S+ConvStem | 0.89 → 2.36 → 6.26 | 1.41 → 2.98 → 7.76 | 1.69 → 3.34 → 10.38 | 0.97 → 2.73 → 8.37 | 1.0 → 2.75 → 8.51 | 1.51 → 3.09 → 8.43 | 0.32 → 1.31 → 4.56 | 0.95 → 2.28 → 6.37 | 0.98 → 2.31 → 6.59 | 1.53 → 3.11 → 8.93 |
| ConvNeXt-B | 0.63 → 1.56 → 4.2 | 0.9 → 1.85 → 4.88 | 0.97 → 2.35 → 7.5 | 0.74 → 1.85 → 5.81 | 0.74 → 1.82 → 5.83 | 1.04 → 2.51 → 5.6 | 0.19 → 0.74 → 2.53 | 0.69 → 1.5 → 4.3 | 0.63 → 1.48 → 4.22 | 0.92 → 2.48 → 6.1 |
| Swin-B | 0.37 → 1.35 → 3.72 | 0.63 → 1.59 → 4.3 | 0.79 → 2.15 → 6.93 | 0.41 → 1.62 → 5.18 | 0.45 → 1.7 → 5.2 | 0.72 → 2.2 → 5.55 | 0.16 → 0.72 → 2.81 | 0.37 → 1.33 → 3.74 | 0.46 → 1.25 → 3.95 | 0.91 → 1.65 → 5.31 |
| ConvNeXt-L | 0.49 → 1.5 → 4.35 | 0.74 → 2.07 → 5.18 | 1.07 → 2.57 → 7.39 | 0.58 → 1.97 → 6.45 | 0.49 → 2.07 → 6.56 | 0.74 → 2.41 → 6.25 | 0.18 → 0.78 → 2.31 | 0.49 → 1.61 → 4.28 | 0.49 → 1.37 → 4.64 | 1.07 → 2.44 → 5.93 |
| ConvV2-L+Swin-L | 2.03 → 3.46 → 6.23 | 2.65 → 4.75 → 7.67 | 1.76 → 3.51 → 7.52 | 2.4 → 4.38 → 9.37 | 2.48 → 4.85 → 9.52 | 2.55 → 4.58 → 8.83 | 0.58 → 1.51 → 3.44 | 2.0 → 3.61 → 6.38 | 1.78 → 3.69 → 6.9 | 2.57 → 4.8 → 9.15 |
| Swin-L | 0.56 → 1.25 → 3.43 | 0.85 → 1.84 → 4.46 | 1.14 → 2.43 → 6.84 | 0.55 → 1.51 → 5.07 | 0.64 → 1.74 → 5.28 | 0.87 → 2.02 → 5.12 | 0.21 → 0.59 → 2.48 | 0.56 → 1.31 → 3.41 | 0.49 → 1.18 → 3.61 | 1.03 → 1.92 → 4.84 |

Table 16: Transfer Attack Performance Across Different Iteration Counts N, 2N, 5N For Epsilon 16/255.

| Model | MI-FGSM | DI-FGSM | TI-FGSM | VMI-FGSM | VNI-FGSM | ADMIX | UAP | SGM | GHOST | LGV |
|---|---|---|---|---|---|---|---|---|---|---|
| ResNet18 | 12.4 → 13.38 → 14.63 | 13.5 → 15.12 → 15.77 | 18.49 → 20.23 → 20.76 | 13.72 → 15.65 → 16.33 | 13.84 → 15.77 → 16.22 | 15.01 → 15.24 → 16.22 | 11.72 → 11.72 → 11.72 | 12.25 → 13.57 → 14.37 | 12.48 → 14.14 → 15.16 | 15.39 → 17.05 → 17.47 |
| ResNet50 | 11.24 → 12.11 → 13.27 | 13.11 → 13.92 → 14.89 | 17.95 → 19.29 → 20.32 | 13.71 → 14.99 → 15.24 | 14.21 → 15.39 → 15.52 | 14.46 → 15.08 → 15.24 | 9.65 → 9.65 → 9.65 | 11.27 → 12.46 → 13.18 | 11.46 → 12.86 → 14.02 | 15.45 → 17.2 → 16.98 |
| WideResNet50.2 | 10.02 → 10.43 → 11.07 | 12.09 → 12.7 → 13.08 | 17.27 → 18.88 → 19.11 | 12.5 → 13.46 → 12.88 | 13.02 → 13.87 → 13.55 | 12.94 → 12.38 → 12.5 | 7.95 → 7.95 → 7.95 | 9.96 → 10.66 → 11.07 | 10.52 → 11.1 → 11.56 | 13.92 → 14.24 → 14.39 |
| ViT-S+ConvStem | 6.26 → 6.59 → 6.73 | 7.76 → 8.32 → 8.15 | 10.38 → 10.38 → 10.54 | 8.37 → 9.01 → 8.46 | 8.51 → 8.93 → 8.6 | 8.43 → 8.43 → 8.09 | 4.56 → 4.56 → 4.56 | 6.37 → 6.68 → 6.87 | 6.59 → 6.82 → 7.04 | 8.93 → 9.18 → 8.54 |
| ConvNeXt-B | 4.2 → 4.12 → 4.3 | 4.88 → 5.17 → 5.12 | 7.5 → 8.47 → 8.13 | 5.81 → 6.04 → 5.54 | 5.83 → 5.94 → 5.49 | 5.6 → 5.3 → 5.28 | 2.53 → 2.53 → 2.53 | 4.3 → 4.28 → 4.22 | 4.22 → 4.35 → 4.57 | 6.1 → 6.04 → 6.12 |
| Swin-B | 3.72 → 3.9 → 4.06 | 4.3 → 4.43 → 5.18 | 6.93 → 7.3 → 7.4 | 5.18 → 5.36 → 5.2 | 5.2 → 5.57 → 5.33 | 5.55 → 5.12 → 4.94 | 2.81 → 2.81 → 2.81 | 3.74 → 3.95 → 4.03 | 3.95 → 4.01 → 4.22 | 5.31 → 5.6 → 5.65 |
| ConvNeXt-L | 4.35 → 4.02 → 4.09 | 5.18 → 5.13 → 5.52 | 7.39 → 7.62 → 7.98 | 6.45 → 6.14 → 5.52 | 6.56 → 6.4 → 5.7 | 6.25 → 5.62 → 5.6 | 2.31 → 2.31 → 2.31 | 4.28 → 4.02 → 4.15 | 4.64 → 4.2 → 4.41 | 5.93 → 6.12 → 5.62 |
| ConvV2-L+Swin-L | 6.23 → 6.41 → 6.51 | 7.67 → 8.31 → 8.43 | 7.52 → 7.99 → 8.06 | 9.37 → 9.2 → 8.61 | 9.52 → 9.55 → 8.98 | 8.83 → 8.34 → 8.19 | 3.44 → 3.44 → 3.44 | 6.38 → 6.36 → 6.6 | 6.9 → 6.73 → 6.73 | 9.15 → 9.03 → 8.9 |
| Swin-L | 3.43 → 3.53 → 3.61 | 4.46 → 4.38 → 4.51 | 6.84 → 7.32 → 6.97 | 5.07 → 4.94 → 4.64 | 5.28 → 5.25 → 4.92 | 5.12 → 4.53 → 4.43 | 2.48 → 2.48 → 2.48 | 3.41 → 3.51 → 3.53 | 3.61 → 3.51 → 3.53 | 4.84 → 4.89 → 4.92 |

rates), while the robust accuracies for GHOST networks is on average 51.54% across the 10 collected models. This confirms that the additional training phase of the LGV models with a high learning rate decreases the robustness of the surrogate models. Consequently, this explains why the performance improvement for LGV models when using a robust surrogate against robust targets is lower compared to the GHOST networks. As a good practice, it is essential to evaluate the robustness of surrogate models prior to the attacking phase. This ensures that any preliminary training phase does not inadvertently weaken the surrogate robustness.

# H    Detailed Tabular Results

We provide the tabular representation corresponding to the results presented in the main paper. Tables 20, 21, 22, 23 represent the success rates for section 3, 4, and 5 with vanilla surrogates and robust surrogates, respectively. These tables display the mean and standard deviation of success rates computed across three random seeds.

Table 17: Query Attack Performance Across Increased Epsilon 4/255, 8/255 and 16/255.

| Model | BASES | TREMBA |
|---|---|---|
| ResNet18 | 2.36 → 5.33 → 13.43 | 4.06 → 11.3 → 37.39 |
| ResNet50 | 2.83 → 5.09 → 13.76 | 3.56 → 9.52 → 32.67 |
| WideResNet50.2 | 2.35 → 5.27 → 12.07 | 3.22 → 9.04 → 31.77 |
| ViT-S+ConvStem | 1.55 → 3.39 → 7.93 | 1.54 → 5.42 → 24.11 |
| ConvNeXt-B | 1.38 → 2.53 → 5.8 | 1.34 → 4.46 → 18.54 |
| Swin-B | 0.96 → 2.26 → 5.2 | 1.14 → 4.11 → 17.17 |
| ConvNeXt-L | 1.16 → 2.7 → 6.04 | 1.47 → 4.43 → 17.21 |
| ConvV2-L+Swin-L | 4.77 → 6.73 → 10.12 | 2.94 → 7.59 → 21.84 |
| Swin-L | 1.26 → 2.18 → 4.79 | 1.25 → 4.02 → 16.41 |

Table 18: Query Attack Performance Across Different Query Counts N, 2N For Epsilon 16/255.

| Model | BASES | TREMBA |
|---|---|---|
| ResNet18 | 13.43 → 13.73 | 37.39 → 43.06 |
| ResNet50 | 13.76 → 14.05 | 32.67 → 37.78 |
| WideResNet50.2 | 12.07 → 12.23 | 31.77 → 36.24 |
| ViT-S+ConvStem | 7.93 → 8.07 | 24.11 → 29.45 |
| ConvNeXt-B | 5.8 → 5.93 | 18.54 → 22.51 |
| Swin-B | 5.2 → 5.2 | 17.17 → 21.44 |
| ConvNeXt-L | 6.04 → 6.17 | 17.21 → 21.02 |
| ConvV2-L+Swin-L | 10.12 → 10.29 | 21.84 → 26.34 |
| Swin-L | 4.79 → 4.99 | 16.41 → 20.46 |

Table 19: Robust accuracies of LGV and GHOST models before the attacking phase.

| | Num 1 | Num 2 | Num 3 | Num 4 | Num 5 | Num 6 | Num 7 | Num 8 | Num 9 | Num 10 | Average |
|---|---|---|---|---|---|---|---|---|---|---|---|
| GHOST | 51.39 | 51.86 | 51.74 | 51.02 | 51.58 | 50.76 | 52.39 | 52.81 | 51.11 | 50.78 | 51.54 |
| LGV | 0 | 0 | 0 | 0 | 0 | 0 | 0 | 0 | 0 | 0 | 0 |

Table 20: Success rates of black-box attacks against one vanilla, and one robust ResNet50 model.

| MODELS | MI-FGSM | DI-FGSM | TI-FGSM | VMI-FGSM | VNI-FGSM | ADMIX | UAP | GHOST | LGV | BASES | TREMBA |
|---|---|---|---|---|---|---|---|---|---|---|---|
| Vanilla ResNet50 | 57.77 ±0.00 | 71.31 ±0.32 | 43.59 ± 0.17 | 66.79 ± 0.07 | 68.00 ± 0.18 | 99.95 ± 0.00 | 8.89 ± 0.20 | 64.53 ± 0.13 | 90.11 ± 0.12 | 81.08 ±0.49 | 89.56 ±0.09 |
| Robust ResNet50 | 1.53 ± 0.00 | 1.94 ± 0.03 | 2.3 ± 0.01 | 1.66 ± 0.03 | 1.85 ± 0.04 | 2.54 ± 0.09 | 0.98 ± 0.10 | 1.68 ±0.03 | 1.33 ±0.01 | 2.83 ±0.04 | 3.56 ±0.06 |

Table 21: Success rates of white-box AutoAttack and black-box attacks against robust targets from Robustbench.

| RANK | MODELS | AUTOATTACK | MI-FGSM | DI-FGSM | TI-FGSM | VMI-FGSM | VNI-FGSM | ADMIX | UAP | SGM | GHOST | LGV | BASES | TREMBA |
|---|---|---|---|---|---|---|---|---|---|---|---|---|---|---|
| 21 | Salman ResNet18 | 50.53 | 1.28 ±0.00 | 1.41 ±0.02 | 2.26 ± 0.04 | 1.24 ± 0.05 | 1.35 ± 0.02 | 1.68 ± 0.06 | 0.86 ± 0.10 | 1.25 ± 0.00 | 1.21 ±0.18 | 2.27 ±0.11 | 2.36 ±0.04 | 4.06 ±0.06 |
| 18 | Salman ResNet50 | 43.84 | 1.68 ± 0.00 | 2.2 ± 0.10 | 2.81 ± 0.07 | 1.89 ± 0.04 | 1.72 ± 0.03 | 2.68 ± 0.13 | 1.02 ± 0.01 | 1.62 ± 0.00 | 1.69 ±0.05 | 2.92 ±0.01 | 2.83 ±0.04 | 3.56 ±0.06 |
| 17 | Salman WideResNet50.2 | 42.87 | 1.34 ± 0.00 | 1.99 ± 0.04 | 2.78 ± 0.10 | 1.48 ± 0.05 | 1.59 ± 0.05 | 2.12 ± 0.08 | 0.93 ± 0.07 | 1.43 ± 0.00 | 1.49 ±0.06 | 2.58 ±0.09 | 2.35 ±0.06 | 3.22 ±0.06 |
| 12 | Sing ViT-S, ConvStem | 34.76 | 0.89 ± 0.00 | 1.41 ± 0.05 | 1.69 ± 0.05 | 0.97 ± 0.04 | 1.00 ± 0.00 | 1.51 ± 0.05 | 0.32 ± 0.06 | 0.95 ± 0.00 | 0.98 ±0.03 | 1.53 ±0.08 | 1.55 ±0.00 | 1.54 ±0.03 |
| 7 | Liu ConvNeXt-B | 28.01 | 0.63 ± 0.00 | 0.9 ± 0.02 | 0.97 ± 0.02 | 0.74 ± 0.02 | 0.74 ± 0.04 | 1.04 ± 0.04 | 0.19 ± 0.04 | 0.69 ± 0.00 | 0.63 ±0.03 | 0.92 ±0.07 | 1.38 ±0.06 | 1.34 ±0.04 |
| 5 | Liu Swin-B | 27.14 | 0.37 ± 0.00 | 0.63 ± 0.03 | 0.79 ± 0.07 | 0.41 ± 0.01 | 0.45 ± 0.02 | 0.72 ± 0.02 | 0.16 ± 0.02 | 0.37 ± 0.00 | 0.46 ±0.07 | 0.91 ±0.03 | 0.96 ±0.02 | 1.14 ±0.03 |
| 3 | Liu ConvNeXt-L | 25.95 | 0.49 ± 0.00 | 0.74 ± 0.09 | 1.07 ± 0.03 | 0.58 ± 0.03 | 0.49 ± 0.02 | 0.74 ± 0.01 | 0.18 ± 0.06 | 0.49 ± 0.00 | 0.49 ±0.09 | 1.07 ±0.03 | 1.16 ±0.00 | 1.47 ±0.01 |
| 2 | Bai ConvNeXtV2-L, Swin-L | 25.48 | 2.03 ± 0.00 | 2.65 ± 0.03 | 1.76 ± 0.05 | 2.40 ± 0.05 | 2.48 ± 0.03 | 2.55 ± 0.05 | 0.58 ± 0.05 | 2.00 ± 0.00 | 1.78 ±0.16 | 2.57 ±0.10 | 4.77 ±0.13 | 2.94 ±0.12 |
| 1 | Liu Swin-L | 25.98 | 0.56 ± 0.00 | 0.85 ± 0.04 | 1.14 ± 0.08 | 0.55 ± 0.01 | 0.64 ± 0.04 | 0.87 ± 0.00 | 0.21 ± 0.04 | 0.56 ± 0.00 | 0.49 ±0.04 | 1.03 ±0.03 | 1.26 ±0.02 | 1.25 ±0.01 |

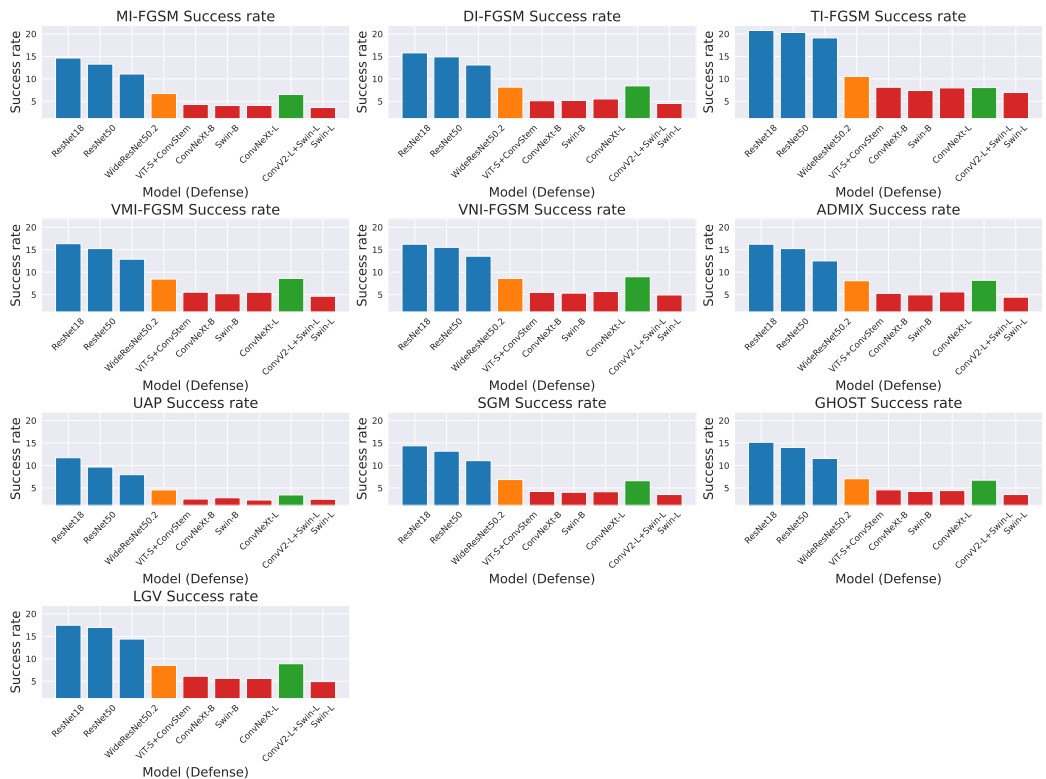

Figure 10: Success rate of blackbox attacks against SoTA defenses for epsilon 16/255 and iteration=5N.

Table 22: Success rates of black-box attacks using vanilla surrogates against a vanilla target and robust targets.

| Models | MI-FGSM | DI-FGSM | TI-FGSM | VMI-FGSM | VNI-FGSM | ADMIX | UAP | GHOST | LGV | BASES | TREMBA |
|---|---|---|---|---|---|---|---|---|---|---|---|
| Standard ResNet50 | $51.62 \pm 0.0$ | $65.95 \pm 0.75$ | $41.5 \pm 0.35$ | $61.22 \pm 0.18$ | $61.10 \pm 0.13$ | $64.75 \pm 0.17$ | $7.11 \pm 0.25$ | $49.36 \pm 0.14$ | $87.39 \pm 0.26$ | $81.08 \pm 0.49$ | $89.56 \pm 0.09$ |
| Salman ResNet18 | $0.98 \pm 0.0$ | $1.51 \pm 0.03$ | $2.04 \pm 0.06$ | $1.29 \pm 0.05$ | $1.35 \pm 0.06$ | $1.88 \pm 0.08$ | $0.83 \pm 0.03$ | $1.13 \pm 0.08$ | $2.17 \pm 0.05$ | $2.36 \pm 0.04$ | $4.06 \pm 0.06$ |
| Liu ConvNeXt-L | $0.6 \pm 0.0$ | $0.92 \pm 0.05$ | $1.19 \pm 0.1$ | $0.7 \pm 0.02$ | $0.8 \pm 0.04$ | $0.99 \pm 0.03$ | $0.18 \pm 0.04$ | $0.48 \pm 0.09$ | $1.05 \pm 0.02$ | $1.16 \pm 0.0$ | $1.47 \pm 0.01$ |
| Liu ConvNeXtV2-L+Swin-L | $2.0 \pm 0.0$ | $2.79 \pm 0.07$ | $2.08 \pm 0.04$ | $2.51 \pm 0.02$ | $2.45 \pm 0.05$ | $2.52 \pm 0.06$ | $0.63 \pm 0.03$ | $1.7 \pm 0.19$ | $2.57 \pm 0.07$ | $4.77 \pm 0.13$ | $2.94 \pm 0.12$ |
| Liu Swin-L | $0.51 \pm 0.0$ | $0.88 \pm 0.03$ | $1.22 \pm 0.05$ | $0.74 \pm 0.02$ | $0.74 \pm 0.04$ | $0.93 \pm 0.05$ | $0.18 \pm 0.0$ | $0.53 \pm 0.03$ | $1.1 \pm 0.11$ | $1.26 \pm 0.02$ | $1.25 \pm 0.01$ |

Table 23: Success rates of black-box attacks using robust surrogates against a vanilla target and robust targets.

| Models | MI-FGSM | DI-FGSM | TI-FGSM | VMI-FGSM | VNI-FGSM | ADMIX | UAP | GHOST | LGV | BASES | TREMBA |
|---|---|---|---|---|---|---|---|---|---|---|---|
| Standard ResNet50 | $15.96 \pm 0.0$ | $15.28 \pm 0.21$ | $8.48 \pm 0.18$ | $16.47 \pm 0.02$ | $16.51 \pm 0.04$ | $15.22 \pm 0.11$ | $0.73 \pm 0.0$ | $17.95 \pm 0.36$ | $81.34 \pm 0.43$ | $17.94 \pm 0.00$ | $67.54 \pm 0.13$ |
| Salman ResNet18 | $12.44 \pm 0.0$ | $13.83 \pm 0.14$ | $12.39 \pm 0.04$ | $12.54 \pm 0.02$ | $12.34 \pm 0.05$ | $14.4 \pm 0.17$ | $0.79 \pm 0.0$ | $14.15 \pm 0.22$ | $2.78 \pm 0.21$ | $11.95 \pm 0.00$ | $10.58 \pm 0.14$ |
| Liu ConvNeXt-L | $9.67 \pm 0.0$ | $9.74 \pm 0.0$ | $7.28 \pm 0.04$ | $9.71 \pm 0.01$ | $9.46 \pm 0.00$ | $9.1 \pm 0.06$ | $0.36 \pm 0.0$ | $9.23 \pm 0.13$ | $1.6 \pm 0.11$ | $12.02 \pm 0.00$ | $4.94 \pm 0.06$ |
| Liu ConvNeXtV2-L+Swin-L | $7.59 \pm 0.0$ | $7.07 \pm 0.09$ | $5.09 \pm 0.04$ | $7.54 \pm 0.03$ | $7.57 \pm 0.00$ | $7.0 \pm 0.02$ | $0.37 \pm 0.0$ | $6.69 \pm 0.08$ | $4.9 \pm 0.06$ | $11.77 \pm 0.01$ | $5.33 \pm 0.11$ |
| Liu Swin-L | $8.94 \pm 0.0$ | $8.72 \pm 0.04$ | $6.78 \pm 0.07$ | $8.95 \pm 0.01$ | $8.89 \pm 0.02$ | $8.21 \pm 0.01$ | $0.31 \pm 0.0$ | $8.16 \pm 0.09$ | $1.57 \pm 0.04$ | $12.55 \pm 0.00$ | $4.4 \pm 0.08$ |

