# OpenReview forum: "RobustBlack: Challenging Black-Box Adversarial Attacks on State-of-the-Art Defenses"
_TMLR — Rejected by TMLR_

### Review · Reviewer_oQbu · 2025-03-12

**Summary Of Contributions:**

This paper proposes a benchmark to study the effectiveness of black-box adversarial attacks on robust models, particularly adversarially trained models and white-box defenses. Experiments on this benchmark show that the threat from black-box attacks can be easily mitigated by white-box defenses; however, utilizing more robust surrogate models leads to more successful attacks.

**Audience:**

Yes

**Broader Impact Concerns:**

This paper has an Acknowledgement section, I am not sure if it violates double-blind.

**Claims And Evidence:**

Yes

**Requested Changes:**

- Including additional results on more recent query-based attacks.
- Although the main focus of this work is white-box defenses, comparing with SoTA defenses for black-box attacks, such as random noise defense and AAA, would provide more insights into the current state of this line of attacks.

"Random noise defense against query-based black-box attacks", NeurIPS 2021

"Adversarial Attack on Attackers: Post-Process to Mitigate Black-Box Score-Based Query Attacks", NeurIPS 2022

"Understanding the Robustness of Randomized Feature Defense Against Query-Based Adversarial Attacks", ICLR 2024

**Strengths And Weaknesses:**

Strengths:
- Propose a novel benchmark with extensive experiments
- Show some interesting insights on the interaction between black-box attacks and white-box defenses.

Weaknesses:
- This benchmark overlooks and misses a lot of query-based attacks, for example Square attack, SimBA, Sign OPT, SignHunt, RayS, SignFlip.
- Insight 1 and 2, while being interesting, lack comparison with more recent query-based attacks. Many works, such as Square attack and RayS, prove that those attacks are effective even against adversarially trained model.

---

> ### Author Response · Authors · 2025-03-26
> **Complete answer to reviewer with empirical comparison of the proposed attacks and the benchmarked attacks**
>
> >W1 + C1: This benchmark overlooks and misses a lot of query-based attacks, for example Square attack, SimBA, Sign OPT, SignHunt, RayS, SignFlip. Including additional results on more recent query-based attacks.
>
> Thank you for pointing our these query attacks. We have included them in the related section of the revised manuscript. We have investigated these attacks except one.
> We could not find an attack called SignFlip for black-box setting, could you please provide us a reference and we will be happy to investigate its implementation in our study.
>
> These attacks have all evaluated ImageNet datasets, using a ResNet50 model for all except SignHunt that only provides an evaluation against InceptionV3. We provide below a comparative study of these approaches based on the figures provided in their respective papers. The table confirms that all these attacks are significantly weaker than the two attacks we included in our study TREMBA and BASES; TREMBA and BASES require five hundred times less queries to achieve perfect success rate.
>
> In addition to including the suggested attacks, we have also added in the revised manuscript a new section (2.5.) to motivate the choice of the attacks included in our empirical study. Our goal is indeed not to survey all transfer attacks, but provide a representative evaluation of the main theories behind adversarial transferability.
>
> | Attack | Succcess Rate | AVG queries |
> | -------- | -------- | -------- |
> | TREMBA     | 98.9     | 7.5     |
> | BASES     | 100     | 1.2     |
> | SquareAttack     | 92.9     | 1100     |
> | SimBA(L2)     | 98.6     | 1665     |
> | Sign OPT (L2)     | 90.0     | 160000     |
> | SignHunt (Inception)     | 99.8     | 578.56|
> | RayS     |    99.8  | 574  |
>
> -----------------
>
> >C2: Although the main focus of this work is white-box defenses, comparing with SoTA defenses for black-box attacks, such as random noise defense and AAA, would provide more insights into the current state of this line of attacks.
>
> Thank you for suggesting the other type of defenses against black-box attacks. We have included them back in the related work and discussed their implications.
>
> We have restricted our evaluation determnistric defenses to adhere strictly to the protocol of RobustBench. They point out for instance that "the classification decision of randomized models may vary over different runs for the same input, hence even the definition of robust accuracy differs from that of deterministic networks."
>
> We would like to point as well that all the suggested defenses were only compared to simple adversarial training [A,C], or weak robustbench defenses (Salman et al. ranked 21 in our Table 1)[B]. The defenses we used are significantly more reliable than these models.
>
> In addition, recent research has also demonstrated that randomness and pre-processing is not a reliable defense, and we can easily design adaptive attacks that actually infer the processing and overcome this defense [D]. Finally, seminal work on good practices of evaluating adversarial robustness [E, F] pointed out that random defenses are safe, and remains less reliable than strong adversarial technique defenses.
>
> Overall, we have included the proposed approaches in the related work, included a discussion about random defenses in the limitation section of our work, and we agree with the reviewer that a further study could investigate how random defenses handle SoTA black-box attacks, and the combination of strong AT approaches and random defenses.
>
>
> [A] "Random noise defense against query-based black-box attacks", NeurIPS 2021
>
> [B] "Adversarial Attack on Attackers: Post-Process to Mitigate Black-Box Score-Based Query Attacks", NeurIPS 2022
>
> [C] "Understanding the Robustness of Randomized Feature Defense Against Query-Based Adversarial Attacks", ICLR 2024
>
> [D] Sitawarin, Chawin, Florian Tramèr, and Nicholas Carlini. "Preprocessors matter! realistic decision-based attacks on machine learning systems." Proceedings of the 40th International Conference on Machine Learning. 2023.
>
> [E] Carlini, Nicholas, et al. "On evaluating adversarial robustness." arXiv preprint arXiv:1902.06705 (2019).
>
> [F] Tramer, Florian, et al. "On adaptive attacks to adversarial example defenses." Advances in neural information processing systems 33 (2020): 1633-1645.

---

### Review · Reviewer_1Kur · 2025-03-13

**Summary Of Contributions:**

This paper presents a comprehensive evaluation framework for black-box adversarial attacks against robust models. The authors show that state-of-the-art black-box attacks, which perform well on standard models, are far less effective when evaluated on adversarially trained models such as those in the RobustBench leaderboard. In addition, the paper demonstrates that robustness achieved through white-box defense methods also translates to improved black-box resistance.

**Audience:**

Yes

**Broader Impact Concerns:**

The paper has a positive broader impact by providing a more realistic assessment of black-box adversarial attacks on robust models. This evaluation can guide both the development of improved attack methods and the design of stronger defense strategies. The insights on surrogate model selection and robustness alignment are likely to inform the deployment of AI systems in scenarios where security against adversarial examples is critical. However, it is important for future research to consider how these findings translate to real-world applications and whether similar trends hold in domains beyond image classification. Overall, the work contributes to a better understanding of the robustness of modern defenses, which is crucial for the safe deployment of machine learning systems.

**Claims And Evidence:**

Yes

**Requested Changes:**

I suggest a few adjustments that would further strengthen the work without being critical for acceptance. First, it would be helpful to improve the clarity in the presentation of hyperparameters and their effect on the attack success rates. A more structured explanation of the attack settings, including additional context for the choice of the perturbation bound

$\|\delta\|_\infty \leq \frac{4}{255}$

would benefit the reader. Second, the authors could include a dedicated section discussing the limitations of the current study, such as the focus on image classification and the need for future work to consider tasks like object detection or segmentation. Third, expanding the mathematical discussion around the transferability of adversarial examples, possibly with further equations or proofs, would add depth to the analysis. Finally, a brief discussion regarding the computational cost and efficiency of the evaluation framework would be valuable. These suggestions are intended to strengthen the work rather than being critical for acceptance.

**Strengths And Weaknesses:**

The paper is strong in its thorough experimental design and the clarity of its evaluation protocols. The presentation of multiple attack methods and defense strategies adds to the paper’s robustness and practical relevance. The use of a large-scale dataset such as ImageNet and a diverse set of models enhances the credibility of the findings. The discussion of the relationship between white-box and black-box robustness is insightful, and the analysis of surrogate model impact is an interesting contribution that can guide future research.

On the other hand, there are areas where the paper could be improved. Some sections, particularly the discussion of hyperparameter choices, could be clarified further. The paper would also benefit from a more detailed discussion of the limitations of the current study, including potential challenges in generalizing the results to other tasks beyond image classification. In addition, the mathematical explanation underlying the observed improvements could be enhanced with additional equations or derivations that detail the transferability mechanisms.

---

> ### Author Response · Authors · 2025-03-26
> **Complete answer for the reviewer with detailed explanations of the computation budget of each attack**
>
> >W1 + C1: First, it would be helpful to improve the clarity in the presentation of hyperparameters and their effect on the attack success rates.
>
> We would like to thank you for your suggestion, we have added a dedicated section in the revision about the impact of hyper-parameters (Appendix D), and provided directly these numbers in the general answer (this was requested by two reviewers). Our new figures confirm that increasing the query budgets or iteration budgets has limited impact on success rate, and increasing the epsilon parameter only shows significant impact for TREMBA attack. This overall is in line with our current main insights. These parameters are the only that could reasonably affect the success rate.
>
> ------------------------------------
>
> >W2 + C2: Second, the authors could include a dedicated section discussing the limitations of the current study, such as the focus on image classification and the need for future work to consider tasks like object detection or segmentation.
>
> Thank you very much for your suggestion, we have indeed scattered the discussion of our limitations through the paper. We have added a dedicated section and discussed extending our work to tasks like segmentation and detection, but also considering weakly-perturbed examples, and the connection to LLM and foundation models testing.
>
> ------------------------------------
>
> >W3 + C3: Third, expanding the mathematical discussion around the transferability of adversarial examples, possibly with further equations or proofs, would add depth to the analysis.
>
> We would like to thank you for this suggestion. Each attack leverages a different hypothesis to explain/motivate its effectiveness. We introduced a new section in the revised manuscript (section 2.5) where we present each hypothesis and connect it to the attacks we have evaluated. If you prefer, in addition to the summary we included, we can also add in the appendix the equations and proofs each approach/hypothesis relied on to support their claims.
>
> -----------------------------
>
> >W4 + C4: Finally, a brief discussion regarding the computational cost and efficiency of the evaluation framework would be valuable.
>
> Thank you for this suggestion. We have added the following table detailing the computational budgets of all studied black-box attacks, quantified as the number of model calls required during the attacking phase per entry. Indeed, each attack uses internal mechanisms to craft their examples. In addition, some attacks require training specific models, for example an ensemble of surrogate for LGV, and a generator for TREMBA. Overall, the search attacks TREMBA and BASES are the most expensive, but also the most effective.
>
> | Attack Algorithm | Number of Model Calls | Explanation |
> |------------------|-----------------------|-------------|
> | MIFGSM  | Iterations (e.g., 10) | MIFGSM iteratively refines the adversarial image over a number of steps, each requiring a model call |
> | DIFGSM  | Iterations (e.g., 10) | DIFGSM performs iterative updates with input diversity applied at each step, requiring a model call per step |
> | TIFGSM  | Iterations (e.g., 10) | TIFGSM performs iterative updates with translation-invariant gradient smoothing at each step, requiring a model call per step |
> | VMIFGSM  | Iterations * (N + 1) (e.g., 60) | VMIFGSM  iterates Steps times. In each iteration, there's one model call for the adversarial image and N model calls for the neighboring images to calculate the gradient variance |
> | VNIFGSM  | Iterations * (N + 1) (e.g., 60) | VNIFGSM, similar to VMIFGSM. In each iteration, there is one model call for the Nesterov and N model calls for the neighboring images to calculate the gradient variance |
> | GHOST  | Iterations (e.g., 10) | GHOST applies Gaussian noise to the skip connection during each iteration (one Ghost model per iteration ), requiring one model call per step |
> | LGV  | Iterations (e.g., 10) | LGV uses an ensemble of models ( lgv_nb_models_epoch * lgv_epochs models), with one model call at each step for each model in the ensemble |
> | Admix | Iterations * Scale * Size (e.g., 150) | In each step, Admix generates Size admixed images, scales them Scale times, and then calculates the loss |
> | UAP  | Max Iterations + Evaluation Calls (preparing the perturbation) , then 1 |  The primary loop runs for max_iter iterations. Additionally, a function is called periodically to evaluate the current perturbation |
> | TREMBA | (num_iters + 1) * sample_size (e.g., 4000) | Tremba iteratively refines the adversarial perturbation within the latent space of a generator network. Each iteration involves generating sample_size perturbations and evaluating their losses |
> | BASES | iterw * n_iters * n_wb (e.g., 2,000) | BASES uses a surrogate ensemble containing n_wb models and performs an attack for n_iters iterations during each query (iterw) |

---

### Review · Reviewer_gvxD · 2025-03-17

**Summary Of Contributions:**

This paper studies the task of image classification under a $\ell_\infty$-norm bounded adversary, i.e., an adversary aiming to attack a classifier $f$ who is allowed to perturb each pixel of the input image $x$ by a small amount $\epsilon$, i.e., $f(x + v)$ where $\||v\||_\infty \leq \epsilon$. This paper studies a black-box adversary, who is only allowed to access the output of the classifier $f(x)$.

In the above setting, this paper primarily studies a collection of black box adversaries proposed in the past few years, who utilize _surrogate_ classifiers $f'$, craft adversarial examples $x'$ for $f'$, and then check whether $x'$ is misclassified by the original classifier $f$, i.e., $f(x') \neq y$.

The main contributions of this paper are empirical evaluations of these existing methods (for $\epsilon = 4/255$) attempting to provide evidence for the following conjectures:

1. Black-Box Attacks perform much worse for adversarially-trained $f$ than for non-adversarially-trained $f$.

2. The success of black-box attacks correlates with the success of white-box attacks.

3. (A) Black-Box attacks utilizing adversarially-trained surrogate classifiers, perform better against adversarially-trained $f$ than non-adversarially-trained $f$. (B) Similarly, Black-Box attacks utilizing non-adversarially-trained surrogate classifiers perform better against non-adversarially-trained $f$ than adversarially-trained $f$.

**Audience:**

No

**Claims And Evidence:**

No

**Requested Changes:**

1. Perform an evaluation of AutoAttack or other white-box attacks to obtain comparisons corresponding to "Insight 1" for the white-box case: Does it show a trend similar to Figure 1? If so, consider reframing the significance of Insight 1.

2. Perform an ablation over several values of $\epsilon$, till the point where the classifier breaks (i.e., the attack success rate is high). Compare this to the attack success rates reported in the black box attack compared to (e.g., MI-FGSM, BASES, TREMBA). Do the conclusions in this manuscript still hold under these $\epsilon$?

3. For a large enough $\epsilon$, perform an ablation over several values of the parameters controlling the computational budget (e.g., #iterations for some of the attacks). Compare this to the attack success rates reported in the attacks compared to.

4. Consider reframing the significance of Insight 1 and Insight 2 in light of the changes 2, 3. Do the observed trends still hold true when the $y$-axis has a much larger range of attack success rates?

5. Compare and contrast the empirical findings obtained in this work to [REF1, REF2] to see how they complement these earlier published findings. Consider rephrasing the significance of Insight 3 in light of these comparisons.

6. For each black box attack evaluated, consider pausing while introducing these methods in the related works section, and provide a description of {the algorithm, how it differs from earlier work, how it improved the state of the art, and shortcomings} for each of these attack methods.

7. Consider editing for the following writing issues in the manuscript:

    1. Abstract: "we establish a framework" -- consider softening language to "we evaluate existing methods against ..."

    2. Abstract: (3) is well studied, consider softening the contribution here.

    3. Intro: "most computer vision APIs ... provide ... categories alongside scores": Microsoft, Amazon, OpenAI do not for many products. Consider rephrasing.

    4. Intro: "... query attacks ... require large amount of queries. BASES ... achieves 95\% ASR wirh a few queries": Paragraph seems self contradictory.

    5. Related Work: "query-based attacks ... have higher success rates than transfer attacks": Reference needed. Also, it seems this should depend on the attack parameters, and how true the logit being provided is to the actual logit, whether there is any gradient masking, etc. Consider elaborating.

    6. Related Work: Defenses are not "against auto attack" specifically. Consider rephrasing section title.

    7. Related Work: "builds on giving" -> "builds on given"

    8. Related Work: our work "focus" -> "focuses" on evaluating

    9. Experiments: Define Attack Success Rate. Additionally, clarify that many of the black box methods studied had experiments on 1000 ImageNet Images specially chosen from a competition dataset or randomly chosen, but this work deviates from this protocol. A similar comment for the value of $\epsilon = 4/255$.

    10. Experiments: Consider moving the {architecture, architecture type} in Table 1 to the Appendix. Additionally, consider using either attack success rate, or robust accuracy consistently throughout the paper, not both.

    11. Experiments: "achieved using indiscriminately Convnets" - there is a typo / missing sentence here.

    12. Acknowledgement: This section should not be included in a double-blind review copy, as it may partially reveal author identities.

**Strengths And Weaknesses:**

## Strengths:

1. The paper attempts to perform an improved evaluation of recent black-box attacks against well known adversarial training based defenses. Such studies are useful to evaluate the effectiveness of latest methods in this fast-developing field of black-box attacks.

## Weaknesses:

### Technical:

1. (Sec.3.2) It is unclear whether "black-box" attacks specifically play any role in conjecture 1 above: Adversarially trained classifiers are simply more robust than their standard counterparts, and any attack is expected to perform worse for an adversarially trained model. The presented evaluation is insufficient to support any deeper conclusion:

    1. The writing in Sec.3.2 suggests that the importance of this observation arises from the _magnitude of the gap_ between the attack success rate on the adversarially-trained classifier v/s the standard classifier. However, there is limited contrast provided for appreciating this: It is unclear whether the effectiveness of white-box attacks differs in the same setting. Apriori, one would expect that a white-box attack would show a very similar trend -- in fact any consistent attack should show a similar trend.

    2. Sec.3.2 does not consider the effect of the crucial parameter $\epsilon$ in drawing this conclusion. Many of the methods evaluated in the paper, e.g., MI-FGSM, TI-FGSM, BASES, evaluate $\epsilon = 16 / 255$, showing much higher attack success rates than $4$% rate shown in this work. An attack strength of $\epsilon = 16 / 255$ is considered a difficult benchmark for black-box attacks, and reducing $\epsilon$ further is expected to render these attacks ineffective, which is what the present paper finds. As a result, it is unclear whether the observed large gap is majorly an artifact of an insufficient evaluation strength.

    3. Sec.3.2 briefly mentions ablating over the hyperparameter of the number of iterations $M$ (aside: the term _budget_ is used in the paper is confusing: it could refer to a computational budget or a perturbation budget), but falls quite short of the level of rigor needed for an empirical contribution, like the present paper. As an example, many of the evaluated black box attacks (e.g., MI-FGSM, TI-FGSM, BASES, etc.), evaluate entire ranges of hyperparameters $\epsilon, M$ to produce a curve of the attack success rate v/s the hyperparameter, to see where exactly the method starts performing well.

    4. These shortcomings in the strengths of the evaluated attacks is also evidenced in {Fig.5 Appendix: BASES, TREMBA} the attack success rates barely move even when the computational budget is quadrupled (aside: it would be good to add the x4 version of the vanilla evaluation bar). This is a sign of the attack strength being insufficient, and immediately calls for increasing the perturbation budget.

2. (Sec.4.2) Generalizing the comment in 1. above, attacks should perform worse against stronger defenses. The weaker a defense, the better the attack success rate of white-box attacks or black-box attacks, and thus, there should be a strong correlation between white-box and black-box success rates. Due to this simple explanation, conjecture 2 should be correct as stated in the paper: however, the limited experimental setting prevents one from obtaining convincing evidence:

    1. Due to the low $\epsilon$ used in this work, the attack success rates(ASRs) are very low. This in turn means that the range of the $y$-axis showing ASRs is too small to discern any convincing trends. This is especially striking in black box success rate vs white box success rate in Fig.3: the paper writes _we observe a clear downward trend_, whereas this reviewer could not see any clear trend on the plot.

    2. _"Insight 2: ... a large model size does not always guarantee better robustness"_: It is unclear whether this observation is an artifact of the very narrow range of success rates, or is an actual lack of correlation observed consistently with attacks having higher $\epsilon$ and $M$.

    3. _"the inclusion of a standard model component causes its performance against black-box attacks to align with that of weaker AutoAttack defenses."_: this is speculation, and not backed by experiment. Additionally, it is unclear what "aligning with" means here.

3. (Sec.5.2) The main conjecture (3) is actually well studied since several years: see [REF1, REF2] as recent references. At a high level, the gradient fields of similarly trained models are similar, and hence attacks tend to transfer between them. In the light of [REF1, REF2], it is important to compare and contrast the empirical findings obtained in this work to see how they complement the earlier findings.


[REF1] Zhang, Yechao, et al. "Why does little robustness help? a further step towards understanding adversarial transferability." IEEE Symposium on Security and Privacy (2024).

[REF2] Springer, Jacob, Melanie Mitchell, and Garrett Kenyon. "A little robustness goes a long way: Leveraging robust features for targeted transfer attacks." Neural Information Processing Systems 34 (2021).

### Writing / Clarity:

The writing in the present manuscript is sometimes speculative, and has a noticeable number of language/grammatical errors, making some parts hard to parse. Some key technical terms like attack success rate are not clearly defined in the text, potentially making the manuscript hard to follow.

### Related Works:

In addition to the references pointed to above, the present description of the black box attacks compared is lacking in rigor -- specifically, the reader is unable to grasp the differences between the methods at a deeper technical level by reading the related work section.  It would be very helpful to the paper to pause while introducing these methods, and provide a description of {the algorithm, how it differs from earlier work, how it improved the state of the art, and shortcomings} for each of these methods.

---

> ### Author Response · Authors · 2025-03-26
> **Answer to requested change #1**
>
> >  Weakness 1.1 + Change 1:The writing in Sec.3.2 suggests that the importance of this observation arises from the magnitude of the gap between the attack success rate on the adversarially-trained classifier v/s the standard classifier. However, there is limited contrast provided for appreciating this: It is unclear whether the effectiveness of white-box attacks differs in the same setting. Apriori, one would expect that a white-box attack would show a very similar trend -- in fact any consistent attack should show a similar trend.
>
> We would like to thank the reviewer for this comment. We agree that it is expected that success rate drops for all attacks, including white-box attacks. What we would like to emphasise here however, is that however new attacks like TREMBA and BASES that were claimed as significant step forward in black-box attacks in terms of success rate and efficiency fall completely short, and are as ineffective as simpler attacks from ten years ago again a straightfoward adversarial training. In white-box setting, we see actually a different trend. While old whitebox attacks are completely ineffective against vanilla adversarial training, recent attacks like AutoAttack on the other hand remains effective. We have updated Figure 1. as requested by adding one "old" white box attack (PGD), and one "more reliable" whitebox attack (AutoAttack). We have updated the insight accordingly.

---

> > ### Author Response · Authors · 2025-03-26
> > **Answer to requested change #5**
> >
> > >W3+C5: Compare and contrast the empirical findings obtained in this work to [REF1, REF2] to see how they complement these earlier published findings. Consider rephrasing the significance of Insight 3 in light of these comparisons.
> >
> > Thank you for suggesting these two publications.
> > We have included a new section 2.5 in the revised manuscript where we address the two suggested related work.
> >
> > Springer et al. (REF2) investigated the link between the robustness of source models and robustness of target models for transferable attacks. Their results show as expected that adversarial examples generated using non-robust networks do not transfer to the adversarially trained networks. However, what they consider as robust is a vanilla adversarial training, which since has been shown to only be slightly robust in Robustbench benchmark and far from optimal. Our study on the other hand leverages state of the art robustitifcation mecvhanisms and explore a large set of architecture for robust surrogates and targets. While our initial results match, our study highlights further insights (eg. using robust sources is actually detrimental when targetting non-robust targets).
> >
> > Zhang et al. (REF1) confirmed the impact of model smoothness and gradient similarity by exploring the impact of weakly adversarial training (i.e., training with mildly perturbed examples). The approach used
> > to train the model, including data augmentation, synthetic data, regularization are considered in some of the models of our study. Our work focuses on robustification mechanism using extreme augmentations. All
> > in all, our results are complementary; as they explore "mildly robust models" and we focus on "extremely robust models".
> >
> > We additionnaly aknowledged these two works in the future work. We have added a new section "Limitations and Future Work" where we suggest exploring the angles proposed by the two references, related to mild perturbations and theoretical bounds.

---

> ### Author Response · Authors · 2025-03-26
> **Answer to requested change #2**
>
> >W1.2 + C2: Sec.3.2 does not consider the effect of the crucial parameter in drawing this conclusion. Many of the methods evaluated in the paper, e.g., MI-FGSM, TI-FGSM, BASES, evaluate , showing much higher attack success rates than \% rate shown in this work. An attack strength of is considered a difficult benchmark for black-box attacks, and reducing further is expected to render these attacks ineffective, which is what the present paper finds. As a result, it is unclear whether the observed large gap is majorly an artifact of an insufficient evaluation strength.
>
> We thank the reviewer for raising this important point. Indeed, attack budget is one paramether where each approach uses its own protocol. TREMBA uses 12/255 as evaluation budget, BASES uses 16/255 as evaluation and Robustbench benchmarked the models against a 4/255 budget. We used Robustbench for most of our study because we would like our work to become the standard for evaluation of black-box, thus aiming for a more ambitious (smaller) attack budget. Indeed, recent black-box attacks under 16/255 all achieve more 99\% success rate, and we would like to discuss what would be needed to go one step further. We agree with you however that the study of the budget is a critical aspect. We provided in appendix C in the initial study an evaluation where the budget of quadrupled, but did not cover all ranges of epsilon. We have updated the manuscript and added a new section 3.3 with additional results with 8/255 and 16/255 budgets. Our results show increasing the epsilon budget as limited impact, and that under epsilon=16/255 icnreasing the iteration budget or query budget is ineffective against robust models, as claimed originally.
>
> Thank you for suggesting this set of experiments and results to confirm our insights; our study is now more generalizable and applicable to hard settings (small epsilon) and easier settings (large epsilon).

---

> ### Author Response · Authors · 2025-03-26
> **Answer to requested changes #3 and #4 (related)**
>
> >W1.3 + C3 + C4: Sec.3.2 briefly mentions ablating over the hyperparameter of the number of iterations (aside: the term budget is used in the paper is confusing: it could refer to a computational budget or a perturbation budget), but falls quite short of the level of rigor needed for an empirical contribution, like the present paper. As an example, many of the evaluated black box attacks (e.g., MI-FGSM, TI-FGSM, BASES, etc.), evaluate entire ranges of hyperparameters to produce a curve of the attack success rate v/s the hyperparameter, to see where exactly the method starts performing well.
>
>
> We would like to thank you for your suggestion of a more extensive study. We would like to confirm that for query attacks the attack budget can refer both to the computation budget (iterations, queries), and to the perturbation budget. We have added a preliminary section to the background with established definitions to the concepts and metrics used in our evaluation.
>
> We also would like to point out that recent research on black-box attacks, for e.g. the defenses suggested by reviewer oQbu (Adversarial At-
> tack on Attackers) also considers the same setting of 4/255 with similar justification about realism and challenging settings study.
>
> For insight 1, we have also provided in sections 3.3 additional results with plots and tables of increased budgets along different axis: Epsilon (Fig.2), iterations (Table 10), and queries (Table 11) and updated the insights accordingly. We also provided the tables with exact numbers in the general answer above.

---

> > ### Author Response · Authors · 2025-03-31
> > **Answer to requested changes #3 and #4 (Results for Insight 2 with increased budgets)**
> >
> > For insight 2, We examined the impact of different defense strategies as attack budget increases with a high epsilon 16/255. The observation that stronger defenses against AutoAttack generally lead to stronger defenses against black-box attacks remains consistent when  extending the iteration budget from N to 2N and 5N with epsilon 16/255 . Similarly, increasing the query budget from N to 2N with epsilon 16/255 does not alter this pattern. The detailed results supporting this conclusion are presented in Appendix F through figures:  Figure 9 illustrates results for an iteration and query budget of 2N, and Figure 10 presents results for an iteration budget of 5N. Additionally, we provided tabular results corresponding to these figures: Table 16 shows the impact of increasing the iteration budget, and Table 18 examines the impact of increasing the query budget.
> >
> > Here are below the new results with increased attack budgets for Insight 2 with a higher epsilon 16/255.
> >
> > - Transfer Attack Performance Across Different Iteration Counts N → 2N → 5N For Epsilon 16/255
> >
> > | Model           | MI-FGSM               | DI-FGSM               | TI-FGSM               | VMI-FGSM              | VNI-FGSM              | ADMIX                 | UAP                   | SGM                   | GHOST                 | LGV                   |
> > |:----------------|:----------------------|:----------------------|:----------------------|:----------------------|:----------------------|:----------------------|:----------------------|:----------------------|:----------------------|:----------------------|
> > | ResNet18        | 12.4 → 13.38 → 14.63  | 13.5 → 15.12 → 15.77  | 18.49 → 20.23 → 20.76 | 13.72 → 15.65 → 16.33 | 13.84 → 15.77 → 16.22 | 15.01 → 15.24 → 16.22 | 11.72 → 11.72 → 11.72 | 12.25 → 13.57 → 14.37 | 12.48 → 14.14 → 15.16 | 15.39 → 17.05 → 17.47 |
> > | ResNet50        | 11.24 → 12.11 → 13.27 | 13.11 → 13.92 → 14.89 | 17.95 → 19.29 → 20.32 | 13.71 → 14.99 → 15.24 | 14.21 → 15.39 → 15.52 | 14.46 → 15.08 → 15.24 | 9.65 → 9.65 → 9.65    | 11.27 → 12.46 → 13.18 | 11.46 → 12.86 → 14.02 | 15.45 → 17.2 → 16.98  |
> > | WideResNet50.2  | 10.02 → 10.43 → 11.07 | 12.09 → 12.7 → 13.08  | 17.27 → 18.88 → 19.11 | 12.5 → 13.46 → 12.88  | 13.02 → 13.87 → 13.55 | 12.94 → 12.38 → 12.5  | 7.95 → 7.95 → 7.95    | 9.96 → 10.66 → 11.07  | 10.52 → 11.1 → 11.56  | 13.92 → 14.24 → 14.39 |
> > | ViT-S+ConvStem  | 6.26 → 6.59 → 6.73    | 7.76 → 8.32 → 8.15    | 10.38 → 10.38 → 10.54 | 8.37 → 9.01 → 8.46    | 8.51 → 8.93 → 8.6     | 8.43 → 8.43 → 8.09    | 4.56 → 4.56 → 4.56    | 6.37 → 6.68 → 6.87    | 6.59 → 6.82 → 7.04    | 8.93 → 9.18 → 8.54    |
> > | ConvNeXt-B      | 4.2 → 4.12 → 4.3      | 4.88 → 5.17 → 5.12    | 7.5 → 8.47 → 8.13     | 5.81 → 6.04 → 5.54    | 5.83 → 5.94 → 5.49    | 5.6 → 5.3 → 5.28      | 2.53 → 2.53 → 2.53    | 4.3 → 4.28 → 4.22     | 4.22 → 4.35 → 4.57    | 6.1 → 6.04 → 6.12     |
> > | Swin-B          | 3.72 → 3.9 → 4.06     | 4.3 → 4.43 → 5.18     | 6.93 → 7.3 → 7.4      | 5.18 → 5.36 → 5.2     | 5.2 → 5.57 → 5.33     | 5.55 → 5.12 → 4.94    | 2.81 → 2.81 → 2.81    | 3.74 → 3.95 → 4.03    | 3.95 → 4.01 → 4.22    | 5.31 → 5.6 → 5.65     |
> > | ConvNeXt-L      | 4.35 → 4.02 → 4.09    | 5.18 → 5.13 → 5.52    | 7.39 → 7.62 → 7.98    | 6.45 → 6.14 → 5.52    | 6.56 → 6.4 → 5.7      | 6.25 → 5.62 → 5.6     | 2.31 → 2.31 → 2.31    | 4.28 → 4.02 → 4.15    | 4.64 → 4.2 → 4.41     | 5.93 → 6.12 → 5.62    |
> > | ConvV2-L+Swin-L | 6.23 → 6.41 → 6.51    | 7.67 → 8.31 → 8.43    | 7.52 → 7.99 → 8.06    | 9.37 → 9.2 → 8.61     | 9.52 → 9.55 → 8.98    | 8.83 → 8.34 → 8.19    | 3.44 → 3.44 → 3.44    | 6.38 → 6.36 → 6.6     | 6.9 → 6.73 → 6.73     | 9.15 → 9.03 → 8.9     |
> > | Swin-L          | 3.43 → 3.53 → 3.61    | 4.46 → 4.38 → 4.51    | 6.84 → 7.32 → 6.97    | 5.07 → 4.94 → 4.64    | 5.28 → 5.25 → 4.92    | 5.12 → 4.53 → 4.43    | 2.48 → 2.48 → 2.48    | 3.41 → 3.51 → 3.53    | 3.61 → 3.51 → 3.53    | 4.84 → 4.89 → 4.92    |
> >
> >
> > - Query Attack Performance Across Different Query Counts N → 2N For Epsilon 16/255
> >
> > | Model           | BASES         | TREMBA        |
> > |:----------------|:--------------|:--------------|
> > | ResNet18        | 13.43 → 13.73 | 37.39 → 43.06 |
> > | ResNet50        | 13.76 → 14.05 | 32.67 → 37.78 |
> > | WideResNet50.2  | 12.07 → 12.23 | 31.77 → 36.24 |
> > | ViT-S+ConvStem  | 7.93 → 8.07   | 24.11 → 29.45 |
> > | ConvNeXt-B      | 5.8 → 5.93    | 18.54 → 22.51 |
> > | Swin-B          | 5.2 → 5.2     | 17.17 → 21.44 |
> > | ConvNeXt-L      | 6.04 → 6.17   | 17.21 → 21.02 |
> > | ConvV2-L+Swin-L | 10.12 → 10.29 | 21.84 → 26.34 |
> > | Swin-L          | 4.79 → 4.99   | 16.41 → 20.46 |

---

> ### Author Response · Authors · 2025-03-26
> **Answer to requested changes #6 and #7**
>
> >C6: For each black box attack evaluated, consider pausing while introducing these methods in the related works section, and provide a description of {the algorithm, how it differs from earlier work, how it improved the state of the art, and shortcomings} for each of these attack methods.
>
> Thank you for your suggestion. Given space limitations, we did not detail the mechanisms of each of the 11 attacks. To explain the differences of the attacks and why we included them, we added in the related work in a new section (2.5) a taxonomy of causes/hypotheses of transferability of adversarial attacks with the reference to one representative paper that suggest this hypothesis. For each hypothesis, we refer to representative attacks that exploit such hypothesis. These attacks are the ones we included in our empirical study. This addition also motivates the choice of these attacks.
>
> We can include detailed explanation of each attack in the appendix if you prefer.
>
> >C7: Consider editing for the following writing issues in the manuscript.
>
> Thank you for suggesting the rephrasing and writing improvement, they indeed improve readability and nuance of our claims. We have included them in the revised submission on OpenReview. We have highlighted in a orange color the changes to the manuscripts following your suggestions.

---

> ### Author Response · Authors · 2025-03-31
> **Answer to requested changes #3 and #4 (Results for Insight 2 with increased epsilon)**
>
> For insight 2, We examined the impact of different defense strategies as epsilon value increases. The observation that stronger defenses against AutoAttack generally lead to stronger defenses against black-box attacks remains consistent when increasing epsilon values from 4/255 to 8/255 and 16/255. The detailed results supporting this conclusion are presented in Appendix F through figures: Figure 7 and Figure 8 correspond to epsilon values of 8/255 and 16/255, respectively. Additionally, we provided tabular results corresponding to these figures: Table 15 and Table 17 present results with an increased perturbation budget covering {4/255, 8/255, 16/255}.
>
> Here are below the new results with increased epsilon for Insight 2.
>
> - Transfer Attack Performance Across Increased Epsilon 8/255 -> 16/255
>
> | Model           | MI-FGSM      | DI-FGSM      | TI-FGSM      | VMI-FGSM     | VNI-FGSM     | ADMIX        | UAP          | SGM          | GHOST        | LGV          |
> |:----------------|:-------------|:-------------|:-------------|:-------------|:-------------|:-------------|:-------------|:-------------|:-------------|:-------------|
> | ResNet18        | 4.08 → 12.4  | 4.61 → 13.5  | 5.63 → 18.49 | 4.39 → 13.72 | 4.5 → 13.84  | 5.52 → 15.01 | 3.44 → 11.72 | 4.08 → 12.25 | 4.12 → 12.48 | 5.82 → 15.39 |
> | ResNet50        | 3.56 → 11.24 | 4.43 → 13.11 | 5.62 → 17.95 | 4.37 → 13.71 | 4.5 → 14.21  | 5.18 → 14.46 | 2.59 → 9.65  | 3.68 → 11.27 | 3.68 → 11.46 | 5.59 → 15.45 |
> | WideResNet50.2  | 3.82 → 10.02 | 4.6 → 12.09  | 6.06 → 17.27 | 4.52 → 12.5  | 4.66 → 13.02 | 5.13 → 12.94 | 2.62 → 7.95  | 3.79 → 9.96  | 3.67 → 10.52 | 5.39 → 13.92 |
> | ViT-S+ConvStem  | 2.36 → 6.26  | 2.98 → 7.76  | 3.34 → 10.38 | 2.73 → 8.37  | 2.75 → 8.51  | 3.09 → 8.43  | 1.31 → 4.56  | 2.28 → 6.37  | 2.31 → 6.59  | 3.11 → 8.93  |
> | ConvNeXt-B      | 1.56 → 4.2   | 1.85 → 4.88  | 2.35 → 7.5   | 1.85 → 5.81  | 1.82 → 5.83  | 2.51 → 5.6   | 0.74 → 2.53  | 1.5 → 4.3    | 1.48 → 4.22  | 2.48 → 6.1   |
> | Swin-B          | 1.35 → 3.72  | 1.59 → 4.3   | 2.15 → 6.93  | 1.62 → 5.18  | 1.7 → 5.2    | 2.2 → 5.55   | 0.72 → 2.81  | 1.33 → 3.74  | 1.25 → 3.95  | 1.65 → 5.31  |
> | ConvNeXt-L      | 1.5 → 4.35   | 2.07 → 5.18  | 2.57 → 7.39  | 1.97 → 6.45  | 2.07 → 6.56  | 2.41 → 6.25  | 0.78 → 2.31  | 1.61 → 4.28  | 1.37 → 4.64  | 2.44 → 5.93  |
> | ConvV2-L+Swin-L | 3.46 → 6.23  | 4.75 → 7.67  | 3.51 → 7.52  | 4.38 → 9.37  | 4.85 → 9.52  | 4.58 → 8.83  | 1.51 → 3.44  | 3.61 → 6.38  | 3.69 → 6.9   | 4.8 → 9.15   |
> | Swin-L          | 1.25 → 3.43  | 1.84 → 4.46  | 2.43 → 6.84  | 1.51 → 5.07  | 1.74 → 5.28  | 2.02 → 5.12  | 0.59 → 2.48  | 1.31 → 3.41  | 1.18 → 3.61  | 1.92 → 4.84  |
>
> - Query Attack Performance Across Increased Epsilon 8/255 -> 16/255
>
> | Model           | BASES        | TREMBA       |
> |:----------------|:-------------|:-------------|
> | ResNet18        | 5.33 → 13.43 | 11.3 → 37.39 |
> | ResNet50        | 5.09 → 13.76 | 9.52 → 32.67 |
> | WideResNet50.2  | 5.27 → 12.07 | 9.04 → 31.77 |
> | ViT-S+ConvStem  | 3.39 → 7.93  | 5.42 → 24.11 |
> | ConvNeXt-B      | 2.53 → 5.8   | 4.46 → 18.54 |
> | Swin-B          | 2.26 → 5.2   | 4.11 → 17.17 |
> | ConvNeXt-L      | 2.7 → 6.04   | 4.43 → 17.21 |
> | ConvV2-L+Swin-L | 6.73 → 10.12 | 7.59 → 21.84 |
> | Swin-L          | 2.18 → 4.79  | 4.02 → 16.41 |

---

### Author Response · Authors · 2025-03-26
**Response to all reviewers**

The authors would like to thank the reviewers for the constructive comments and suggestions. Overall the reviewers agree on the importance of the problem we are investigated and the extenseiveness of our study. The main criticism is the need for additional ablation studies regarding the impact of larger perturbation budgets epsilon, and whether our insights stand in this setting.

We first would like to insist that the protocol we followed (using epsilon 4/255) was to implemented to follow the best practices and shared standard of evaluating adversarial robustness, following Robustbench. The fact that most publications use much larger perturbation budgets (8/255 or 16/255) was mostly used to boast success rate results. We believe the study of adversarial robustness under such high budgets is a solved problem with attacks like BASES achieveing 99\%+ succcess rate with a few queries.


In the revised manuscript (resubmitted in OpenReview), all the comments from reviewers are considered and accommodated, including a rework of the related work methodology for better readability, and additional proof-reading. In particular, we have added new results with larger budgets, and new baselines. We only included in the main revised paper a subset of the figures and results of space limitation, but we provide in the paper's appendices a more detailed analysis and discussion of the results. In the revised manuscript, we used color coding to showcase:
- In magenta the new content to the manuscript.
- In orange, the content that was reformulated according to the reviews.

Here are below the new results with increased budgets for Insight 1. For each table, we provide the success rate on the vanilla model then "->" followed by the success rate on the robust model:


- Impact of increasing perturbation budget whithout modifying the other parameters:

Attack | Epsilon 4/255 | Epsilon 8/255 | Epsilon 16/255 |
| -------- | -------- |-------- |-------- |
| MI-FGSM     | 57.77 -> 1.53  | 77.33 -> 3.53 | 88.79 -> 10.52 |
| DI-FGSM     | 71.31 ->  1.94  | 87.07 ->  4.18 | 94.55 ->  12.11|
| TI-FGSM     | 43.59 ->   2.30| 66.35 ->  4.68 | 83.53 ->  15.61 |
| VMI-FGSM    | 66.79 ->   1.66| 87.59 ->  3.97 | 94.79 ->  12.74|
| VNI-FGSM    | 68.00 ->   1.85| 90.10 ->  4.09 | 96.94 ->  13.21 |
| ADMIX       | 77.90 ->   2.54| 93.04 ->   4.68| 98.28 ->  14.24|
| UAP         |  8.89 ->   0.98| 25.70 ->  2.37 | 70.78 ->   9.93|
| GHOST       | 64.53 ->   1.68| 83.87 ->   3.59| 94.40 ->  11.27|
| LGV         | 90.11 ->   1.33| 98.07 ->  5.25 | 99.66 ->  15.16|
| BASES       | 81.08 ->   2.83| 96.06 ->   5.09| 99.56 ->  13.76|
| TREMBA      | 89.56 ->   3.56| 99.45 ->   9.52| 99.84 ->  32.67|

- Impact of increasing computation budget using the largest epsilon budget (16/255):

Attack      | Iteration=N | Iteration=2N | Iteration=5N |
 --------   | --------    |--------      |--------      |
MI-FGSM    | 88.79 -> 10.52     | 91.63 -> 11.61      | 93.79 ->  12.67    |
DI-FGSM    | 94.55 -> 12.11     | 97.32 -> 13.30      | 99.17 ->  14.38    |
TI-FGSM    | 83.53 -> 15.61     | 89.31 -> 17.61      | 93.45 ->  18.69    |
VMI-FGSM   | 94.79 -> 12.74     | 97.73 -> 14.30     | 99.06 ->   14.82  |
VNI-FGSM   | 96.94 -> 13.21     | 98.62 -> 14.67      | 99.45 ->  14.98    |
ADMIX      | 98.28 -> 14.24     | 98.90 -> 14.58     | 99.43 ->   15.02   |
UAP        | 70.87 -> 9.93     | 70.78 -> 9.93     | 72.38 ->   10.02   |
GHOST      | 94.40 -> 11.27     | 96.48 -> 12.21     | 98.41 ->  13.39    |
LGV        | 99.66 -> 15.16     | 99.92 -> 16.61     | 99.97 ->  16.39    |

| Model   | Attack  | Query=N | Query=2N |
|---------|-------- |-------- |--------- |
| Vanilla | BASES  | 99.56   | 99.66    |
| Vanilla | TREMBA | 99.84   | 99.90    |
| Robust  | BASES  | 13.76   | 14.05    |
| Robust  | TREMBA | 32.67   | 37.79    |


Overall, our new results confirm the claims of our publication:

- Simple adversarial training is sufficient to counter recent SoTA blackbox attacks
- Increasing computation budget (iterations and queries) have limited impact against robustified models
- TREMBA is the only attacks where increasing epsilon budgets leads a signficant improvement of success rate.

---

### Decision · Action_Editor_syqf · 2025-05-06

**Recommendation:** Reject

**Comment:**

There are outstanding issues that need to be addressed, as described above. I have hesitation to recommend “Accept with minor revision”, as the further experiments should better be considered as major revision. Moreover, the findings of the new experiments may weaken the claims made in the paper. As such, it is more appropriate to review a new submission incorporating all the changes before making a decision, if the authors choose to pursue it.

**Audience:**

The topic studied in this paper would be relevant to researchers in the TMLR community interested in the adversarial robustness of machine learning models.

**Claims And Evidence:**

This paper challenges the sufficiency of existing studies on black-box adversarial attacks as they have not been benchmarked against more recent robust models with stronger defense mechanisms. Through empirical evaluation, the authors show that these models are actually robust against such black-box attacks.

The reviewers raised a number of issues regarding the study. We thank the authors for their effort to address the comments and concerns and revise their original manuscript. Nevertheless, there still exist some outstanding issues in the latest version of the paper. They need to be addressed thoroughly before the paper is ready for publication.

For example, some concerns raised by reviewer gvxD have not been addressed satisfactorily:

* Weakness 1: Figure 1 has not included PGD and AutoAttack and Insight 1 has not been updated accordingly, as claimed by the authors.

* Section 3.3: The new results show that the budget 16/255 should really be the regime for testing the conjectures. Since humans can easily classify images under even higher budgets, 16/255 is actually quite a reasonable budget to consider by the adversarial learning community.

* Weakness 2: The concerns are legitimate but they have not been addressed in the latest version.

In addition, the authors argue that query-based attacks require more queries and thus it is not necessary to evaluate them in the benchmark. However, there is a clear distinction between transfer-based and query-based attacks in the implementation: while transfer-based attacks find adversarial examples of the **surrogate models**, query-based attacks directly find adversarial examples of the **victim model**. If the victim model is more robust, transfer-based attacks will not generalize well, but it does not mean that query-based attacks will fail. It is important to also cover methods that focus on optimizing the searching process without surrogate models.

**Resubmission Of Major Revision:**

The authors may consider submitting a major revision at a later time.